# Recovering Hidden Reward in Diffusion-Based Policies

**Yanbiao Ji** [1]   **Qiuchang Li** [1]   **Yuting Hu** [1]   **Shaokai Wu** [1]   **Wenyuan Xie** [1]   **Guodong Zhang** [1]   **Qicheng He** [1]
**Deyi Ji** [2]   **Yue Ding** [1]   **Hongtao Lu** [1]

## Abstract

This paper introduces ENERGYFLOW, a framework that unifies generative action modeling with inverse reinforcement learning by parameterizing a scalar energy function whose gradient is the denoising field. We establish that under maximum-entropy optimality, the score function learned via denoising score matching recovers the gradient of the expert's soft Q-function, enabling reward extraction without adversarial training. Formally, we prove that constraining the learned field to be conservative reduces hypothesis complexity and tightens out-of-distribution generalization bounds. We further characterize the identifiability of recovered rewards and bound how score estimation errors propagate to action preferences. Empirically, ENERGYFLOW achieves state-of-the-art imitation performance on various manipulation tasks while providing an effective reward signal for downstream reinforcement learning that outperforms both adversarial IRL methods and likelihood-based alternatives. These results show that the structural constraints required for valid reward extraction simultaneously serve as beneficial inductive biases for policy generalization. The code is available at https://github.com/sotaagi/EnergyFlow.

## 1. Introduction

Diffusion-based policies (Chi et al., 2023; Zhang et al., 2025b; Reuss et al., 2024) have become a promising paradigm for embodied agents to learn manipulation skills from expert demonstrations. These methods learn to generate actions by iteratively denoising corrupted samples conditioned on the current state. Due to their capacity to

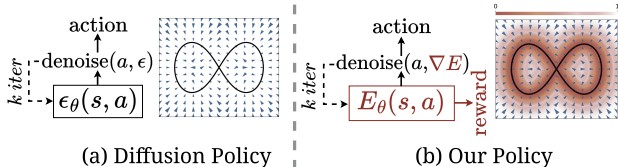

*Figure 1.* **Comparison between Diffusion Policy and ENER-GYFLOW.** (a) Conventional diffusion policies predict noise $\epsilon_\theta(\boldsymbol{s}, \boldsymbol{a})$ for iterative denoising but lack an explicit energy representation. (b) ENERGYFLOW parameterizes an energy function $E_\theta(\boldsymbol{s}, \boldsymbol{a})$ and performs denoising via its gradient $\nabla E_\theta$, enabling action generation as well as outputting reward signals.

model complex, multi-modal distributions, diffusion policies are particularly well-suited for capturing diverse expert behaviors (Chi et al., 2023).

Despite this expressiveness, diffusion policies are typically trained under the behavior cloning (BC) objective (Torabi et al., 2018). They imitate trajectories without explicitly modeling why an action is desirable, *i.e.*, the underlying intent or task preference that makes some behaviors succeed (Hayes & Shah, 2017). In practice, this can limit robustness and extrapolation. When test-time situations deviate from the demonstration distribution, matching action likelihood alone may not provide a reliable signal for action selection (Acero & Li, 2024).

A natural way to model intent is through reward-based Reinforcement Learning (RL). For embodied agents, reward-driven behavior has been widely regarded as important in terms of governing complex cognitive abilities such as perception, imitation, and learning (Lu et al., 2025). This has motivated combining diffusion policies with reinforcement learning, aiming to improve adaptation beyond pure BC (Ada et al., 2024; Ren et al., 2025). However, applying RL in real robotic settings remains challenging, in large part due to the need for careful reward design and tuning (Ye et al., 2024). While inverse reinforcement learning (IRL) methods (Ramachandran & Amir, 2007; Ziebart et al., 2008) can learn rewards from demonstrations, they often bring substantial computational overhead and may suffer from training instabilities (Nijkamp et al., 2022; Du et al., 2021).

We propose to exploit the reward signal that is already implicit in diffusion-based imitation. Motivated by connec-

[1]Shanghai Jiao Tong University [2]KOKONI 3D, Moxin Technology. Correspondence to: Yue Ding <dingyue@sjtu.edu.cn>, Hongtao Lu <htlu@sjtu.edu.cn>.

*Proceedings of the $43^{rd}$ International Conference on Machine Learning*, Seoul, South Korea. PMLR 306, 2026. Copyright 2026 by the author(s).

tions between diffusion models and energy-based modeling (Wang & Du, 2025; Balcerak et al., 2025), we parameterize a scalar energy function over observation–action pairs and train it through a denoising score matching process. The resulting energy landscape both (i) induces a generative vector field for action sampling via its gradient and (ii) provides a reward signal aligned with the Boltzmann form as in maximum-entropy IRL. Figure 1 compares standard diffusion policies, which learn a denoising vector field, with our approach, which also learns the underlying energy function.

Our contributions are as follows:

- We propose ENERGYFLOW, which parameterizes a scalar energy function $E_\theta(o, a)$ and derives the generative vector field from its action-gradient $\nabla_a E_\theta(o, a)$. This enforces integrability by construction and yields complete probability-flow ordinary differential equation (ODE) derivations that connect training and sampling.

- We prove that the integrability constraint acts as implicit regularization, reducing hypothesis complexity and tightening generalization bounds. We further bound how score matching error propagates to recovered action preferences when using the learned energy as a reward signal.

- Through extensive empirical experiments, we show that (i) the learned energy provides an effective shaping signal for downstream RL, with gains attributable to the energy-based extraction method; and (ii) enforcing integrability improves out-of-distribution generalization relative to unconstrained flow policies.

## 2. Preliminaries

**Denoising Score Matching.** Score matching (Hyvärinen, 2005) aims to estimate the score function $\nabla_x \log p(x)$ of a data distribution. Denoising score matching (Vincent, 2011) provides a tractable objective by perturbing data with noise and learning to denoise the corrupted samples. Formally, given a noise-perturbation kernel $q_\sigma(\tilde{x}|x_0) = \mathcal{N}(\tilde{x}; x_0, \sigma^2 I)$, the denoising score matching objective is:

$$\mathbb{E}_{q_\sigma(\tilde{x}|x_0)p(x_0)} \left[ \|\mathcal{S}_\theta(\tilde{x}, \sigma) - \nabla_{\tilde{x}} \log q_\sigma(\tilde{x}|x_0)\|^2 \right], \quad (1)$$

which is equivalent to explicit score matching up to a constant (Vincent, 2011). Since $\nabla_{\tilde{x}} \log q_\sigma(\tilde{x}|x_0) = -(\tilde{x} - x_0)/\sigma^2 = -\varepsilon/\sigma$, the objective reduces to predicting the scaled noise direction.

**Score-Based Generative Models.** Score-based generative models (Song et al., 2021) extend denoising score matching across noise scales. The forward process adds noise according to a schedule $\sigma(t)$ for $t \in [0, T]$:

$$x_t = x_0 + \sigma(t)\varepsilon, \quad \varepsilon \sim \mathcal{N}(0, I), \quad (2)$$

where $\sigma(t)$ is monotonically increasing with $\sigma(0) \approx 0$. A noise-conditional score network $\mathcal{S}_\theta(x, t)$ is trained to approximate $\nabla_x \log p_t(x)$ via the multi-scale objective:

$$\mathcal{L}(\theta) = \mathbb{E}_{t \sim \mathcal{U}[0,T], x_0, \varepsilon} \left[ \lambda(t) \left\| \mathcal{S}_\theta(x_t, t) + \frac{\varepsilon}{\sigma(t)} \right\|^2 \right], \quad (3)$$

where $\lambda(t) = \sigma^2(t)$ ensures uniform contribution across noise levels. Sampling proceeds by integrating the probability-flow ODE from $t = T$ to $t \approx 0$:

$$\frac{dx}{dt} = -\frac{1}{2} \frac{d[\sigma^2(t)]}{dt} \mathcal{S}_\theta(x, t). \quad (4)$$

**Diffusion-Based Policies.** Diffusion-based policies (Chi et al., 2023; Zhang et al., 2025b) represent the policy $\pi_\theta(a|s)$ as a conditional score-based model. The model learns a noise-conditional score network $\mathcal{S}_\theta(a_t, s, t)$ that approximates $\nabla_{a_t} \log p_t(a_t|s)$, trained by minimizing the noise prediction error:

$$\mathcal{L}_{\text{BC}}(\theta) = \mathbb{E}_{t, \varepsilon} \left[ \lambda(t) \left\| \mathcal{S}_\theta(a_t, s, t) + \frac{\varepsilon}{\sigma(t)} \right\|^2 \right], \quad (5)$$

where $a_t = a_0 + \sigma(t)\varepsilon$. At inference, actions are generated by sampling $a_T \sim \mathcal{N}(0, \sigma^2(T)I)$ and integrating the probability-flow ODE Eq. (4) conditioned on $s$.

## 3. Theoretical Analysis

Our goal is to unify generative score matching and inverse reinforcement learning (IRL). In this section, we establish that the score function learned by diffusion models is not merely a sampling mechanism, but an implicit representation of the expert's reward structure.

### 3.1. Equivalence Between Scores and Reward Gradients

Standard diffusion models estimate the score function $\nabla_a \log p_t(a|s)$ to generate data. We first demonstrate that for an optimal embodied agent, this score function already contains the underlying reward function gradients.

**Assumption 3.1** (Maximum Entropy Optimality). *The expert policy $\pi_E(a|s)$ is optimal with respect to the soft Q-function $Q^*(s, a)$ under the Maximum Entropy principle (Ziebart et al., 2008). The policy takes the form of a Boltzmann distribution:*

$$\pi_E(a|s) = \frac{1}{Z(s)} \exp \left( \frac{Q^*(s, a)}{\alpha} \right),$$
$$\text{and } Z(s) = \int \exp \left( \frac{Q^*(s, a)}{\alpha} \right) da, \quad (6)$$

*where $\alpha$ is the temperature parameter and $Q^*(s, a)$ is the optimal soft action-value function incorporating both immediate rewards and future discounted returns.*

*Remark* 3.2 (Scope of the Assumption). In the sequential MDP setting, the partition function satisfies $\log Z(\boldsymbol{s}) = V^*(\boldsymbol{s})/\alpha$, where $V^*$ is the optimal soft value function. Thus $\log \pi_E(\boldsymbol{a}|\boldsymbol{s}) = (Q^*(\boldsymbol{s},\boldsymbol{a}) - V^*(\boldsymbol{s}))/\alpha = A^{\text{soft}}(\boldsymbol{s},\boldsymbol{a})/\alpha$, where $A^{\text{soft}}$ is the soft advantage. Our analysis recovers the soft advantage (or equivalently, the soft Q-function up to state-dependent terms) from demonstrations.

Under this assumption, the relationship between the data distribution and the soft Q-function is linear in log-space. By taking the gradient with respect to the action $\boldsymbol{a}$, we eliminate the intractable partition function $Z(\boldsymbol{s})$, establishing a direct link between the score and the Q-function gradient.

**Theorem 3.3** (Score-Reward Equivalence). *Let $\mathcal{S}^*(\boldsymbol{a},\boldsymbol{s}) := \nabla_{\boldsymbol{a}} \log \pi_E(\boldsymbol{a}|\boldsymbol{s})$ be the true score function of the expert policy. Under Assumption 3.1, the gradient of the expert's soft Q-function is proportional to the score:*

$$\nabla_{\boldsymbol{a}} Q^*(\boldsymbol{s},\boldsymbol{a}) = \alpha \cdot \mathcal{S}^*(\boldsymbol{a},\boldsymbol{s}). \tag{7}$$

*Consequently, if a parameterized energy function $E_\phi(\boldsymbol{a},\boldsymbol{s})$ is trained such that $-\nabla_{\boldsymbol{a}} E_\phi \approx \mathcal{S}^*$, then $E_\phi$ recovers the soft Q-function up to a state-dependent constant:*

$$E_\phi(\boldsymbol{a},\boldsymbol{s}) = -\frac{Q^*(\boldsymbol{s},\boldsymbol{a})}{\alpha} + c(\boldsymbol{s}). \tag{8}$$

*Proof.* Taking the logarithm of Eq. (6) yields $\log \pi_E(\boldsymbol{a}|\boldsymbol{s}) = \frac{1}{\alpha} Q^*(\boldsymbol{s},\boldsymbol{a}) - \log Z(\boldsymbol{s})$. Since $Z(\boldsymbol{s})$ depends only on state $\boldsymbol{s}$, $\nabla_{\boldsymbol{a}} \log Z(\boldsymbol{s}) = 0$. Differentiating both sides with respect to $\boldsymbol{a}$ immediately yields Eq. (7). Integrating both sides with respect to $\boldsymbol{a}$ along any path yields Eq. (8), where $c(\boldsymbol{s})$ is the integration constant. □

**Corollary 3.4** (Connection to Soft Advantage). *Under Assumption 3.1, the learned energy satisfies:*

$$E_\phi(\boldsymbol{a},\boldsymbol{s}) = -\frac{A^{soft}(\boldsymbol{s},\boldsymbol{a})}{\alpha} + c'(\boldsymbol{s}), \tag{9}$$

*where $A^{soft}(\boldsymbol{s},\boldsymbol{a}) = Q^*(\boldsymbol{s},\boldsymbol{a}) - V^*(\boldsymbol{s})$ is the soft advantage and $c'(\boldsymbol{s}) = c(\boldsymbol{s}) + V^*(\boldsymbol{s})/\alpha$.*

This theorem suggests that score matching can substitute for the unstable min-max optimization typical of adversarial IRL. However, Eq. (7) only holds if the learned score field is actually the gradient of a *scalar function*. This leads to a need for proper structural constraints.

### 3.2. Enforcing Conservative Field

While Theorem 3.3 establishes that a reward gradient is a score, the converse is not automatically true for approximated functions. A generic neural network outputting a vector field may not be the gradient of any scalar field.

**Definition 3.5** (Conservative Vector Field). A vector field $V : \mathbb{R}^d \to \mathbb{R}^d$ is *conservative* (or integrable) if there exists a scalar potential $\Psi$ such that $V = \nabla \Psi$. A necessary

condition is that the Jacobian is symmetric ($\nabla \times V = 0$), implying path independence.

If a learned score field $\mathcal{S}_\phi$ is not conservative, the implied "reward" becomes ill-defined. Specifically, a non-conservative field induces *cyclic preferences* (e.g., $\boldsymbol{a}_1 \succ \boldsymbol{a}_2 \succ \boldsymbol{a}_3 \succ \boldsymbol{a}_1$), violating the transitivity axiom of rational decision-making (Jiang et al., 2011). To prevent this, we must strictly restrict our hypothesis space to conservative fields. This is achieved by parameterizing a scalar energy network $E_\phi$ and defining the score as $\mathcal{S}_\phi = -\nabla_{\boldsymbol{a}} E_\phi$.

Beyond ensuring theoretical validity, this restriction acts as a powerful inductive bias for generalization.

**Theorem 3.6** (Complexity Reduction via Conservative Constraints). *Let $\phi : \mathbb{R}^{in} \to \mathbb{R}^k$ be a neural feature representation with bounded feature norm $\sup_{\boldsymbol{x}} \|\phi(\boldsymbol{x})\|_2 \leq B$, bounded Jacobian Frobenius norm $\sup_{\boldsymbol{x}} \|J_\phi(\boldsymbol{x})\|_F \leq L$, and bounded weight matrix norm $\sup \|\boldsymbol{W}\| \leq \Lambda$ for the linear map. Let $\mathcal{F}_{unc}$ be the class of arbitrary linear vector fields over $\phi$, and $\mathcal{F}_{cons}$ be the class of conservative vector fields (gradients of potentials over $\phi$). The Empirical Rademacher complexity of the conservative class is strictly tighter with respect to the output dimension $d$:*

$$\hat{\mathfrak{R}}_S(\mathcal{F}_{unc}) \leq \frac{\Lambda B\sqrt{d}}{\sqrt{n}}, \quad \hat{\mathfrak{R}}_S(\mathcal{F}_{cons}) \leq \frac{\Lambda L}{\sqrt{n}}. \tag{10}$$

*For high-dimensional action spaces where $d$ is large, provided the representation is smooth ($L \ll B\sqrt{d}$), we have $\hat{\mathfrak{R}}_S(\mathcal{F}_{cons}) \ll \hat{\mathfrak{R}}_S(\mathcal{F}_{unc})$.*

□ *Proof in Appendix A.1.*

*Remark* 3.7 (Applicability to Deep Architectures). While Theorem 3.6 formally bounds the final linear readout, its assumptions are satisfied by deep neural networks under standard Lipschitz constraints. For a deep network $\phi$, the Jacobian norm $L$ is bounded by the product of the spectral norms of individual weight matrices (Bartlett et al., 2017). In practice, training techniques such as weight decay and spectral normalization strictly control these norms to prevent exploding gradients, ensuring finite $\Lambda$ and $L$.

### 3.3. OOD Generalization

By enforcing a conservative field, we also impose a global structural constraint: the learned field must remain the gradient of a scalar potential even in unseen regions. This forces the model to extrapolate the shape of the energy landscape rather than fitting arbitrary vector directions, effectively coupling the prediction errors across dimensions.

**Lemma 3.8** (OOD Generalization). *Let $\mathcal{D}_S$ be the source training distribution and $\mathcal{D}_T$ be a target (OOD) distribution. Let $h^* \in \mathcal{F}_{cons}$ be the ground truth conservative field. Assume that all hypotheses in $\mathcal{F}_{cons}$ and $\mathcal{F}_{unc}$ are uniformly*

bounded by $M > 0$ (i.e., $\sup_{\boldsymbol{x}} \|f(\boldsymbol{x})\|_2 \leq M$ for all $f$ in the hypothesis class). For any learned hypothesis $f$, let the risk be $\mathcal{R}_{\mathcal{D}}(f) = \mathbb{E}_{\boldsymbol{x} \sim \mathcal{D}}[\|f(\boldsymbol{x}) - h^*(\boldsymbol{x})\|^2]$. The risk on the target domain for the conservative estimator satisfies, with probability at least $1 - \delta$:

$$
\begin{aligned}
\mathcal{R}_{\mathcal{D}_T}(\hat{f}_{cons}) \leq & \hat{\mathcal{R}}_S(\hat{f}_{cons}) \\
& + \frac{1}{2} d_{\mathcal{H}\Delta\mathcal{H}}(\mathcal{D}_S, \mathcal{D}_T) + \mathcal{O}\left(\frac{M\Lambda L}{\sqrt{n}}\right),
\end{aligned} \tag{11}
$$

whereas for the unconstrained estimator $\hat{f}_{unc}$, the complexity term scales with $\mathcal{O}(M\Lambda B\sqrt{d}/\sqrt{n})$. Here, $d_{\mathcal{H}\Delta\mathcal{H}}$ is the discrepancy distance between domains and $\hat{\mathcal{R}}_S$ denotes the empirical source risk.

☐ *Proof in Appendix A.2.*

Lemma 3.8 implies that as the dimensionality $d$ of the action space increases, the upper bound on the OOD error for unconstrained fields grows with $\sqrt{d}$, while the bound for conservative fields remains controlled by the smoothness $L$.

### 3.4. Identifiability and Within-State Reward Shaping

Having established that we can recover a valid reward gradient $\nabla_{\boldsymbol{a}} Q^*$, we must determine if this uniquely identifies the Q-function. Integrating Eq. (7) with respect to $\boldsymbol{a}$ yields:

$$
Q^*(\boldsymbol{s}, \boldsymbol{a}) = -\alpha E_\phi(\boldsymbol{a}, \boldsymbol{s}) + c(\boldsymbol{s}), \tag{12}
$$

where $c(\boldsymbol{s})$ is an unknown state-dependent integration constant. This represents a fundamental limit of learning from demonstrations: we observe which actions are preferred *at* a state, but not how good the state is globally.

**Proposition 3.9** (Within-State Action Ranking). *The learned energy provides exact within-state action rankings:*

1. **Within-state ranking is exact.** *For any fixed state $\boldsymbol{s}$, the action with lowest energy is the expert's most preferred action:* $\arg\min_{\boldsymbol{a}} E_\phi(\boldsymbol{a}, \boldsymbol{s}) = \arg\max_{\boldsymbol{a}} Q^*(\boldsymbol{s}, \boldsymbol{a})$.

2. **Cross-state comparison is ambiguous.** *The difference $E_\phi(\boldsymbol{a}, \boldsymbol{s}) - E_\phi(\boldsymbol{a}', \boldsymbol{s}')$ includes the unknown quantity $c(\boldsymbol{s}) - c(\boldsymbol{s}')$.*

☐ *Proof in Appendix A.3.*

*Remark* 3.10 (State Ambiguity). The recovered reward $\hat{r}(\boldsymbol{s}, \boldsymbol{a}) = -\alpha E_\phi(\boldsymbol{a}, \boldsymbol{s})$ differs from the true soft Q-function by a state-dependent offset $c(\boldsymbol{s})$. In the specific case where $c(\boldsymbol{s})$ takes the form required by potential-based reward shaping (PBRS) (Ng et al., 1999), *i.e.*, it can be expressed as a potential difference $\gamma\Phi(\boldsymbol{s}') - \Phi(\boldsymbol{s})$ over transitions, the optimal policy is provably preserved. In general, however, a state-only offset does *not* satisfy the PBRS form and may alter the optimal policy in sequential settings. Nevertheless,

---

**Algorithm 1** ENERGYFLOW Training

**Require:** Expert dataset $\mathcal{D} = \{(\boldsymbol{s}_i, \boldsymbol{a}_i)\}$, energy network $E_\phi$, noise schedule $\sigma(t) = \sigma_{\min}^{1-t/T} \sigma_{\max}^{t/T}$
1: **for** each training iteration **do**
2:      Sample batch $(\boldsymbol{s}, \boldsymbol{a}) \sim \mathcal{D}$
3:      Sample $t \sim \mathcal{U}[0, T]$, $\boldsymbol{\varepsilon} \sim \mathcal{N}(\boldsymbol{0}, \boldsymbol{I})$
4:      Form noisy action $\boldsymbol{a}_t = \boldsymbol{a} + \sigma(t)\boldsymbol{\varepsilon}$
5:      Compute $\mathcal{S}_\phi(\boldsymbol{a}_t, \boldsymbol{s}, t) = -\nabla_{\boldsymbol{a}_t} E_\phi(\boldsymbol{a}_t, \boldsymbol{s}, t)$ via autodiff
6:      Compute loss $\mathcal{L} = \sigma^2(t)\|\mathcal{S}_\phi(\boldsymbol{a}_t, \boldsymbol{s}, t) + \boldsymbol{\varepsilon}/\sigma(t)\|^2$
7:      Update $\phi$ by gradient descent on $\mathcal{L}$
8: **end for**

---

for *within-state action selection* (which is the primary use case for our shaping signal in downstream RL), the offset $c(\boldsymbol{s})$ is irrelevant since it cancels when comparing actions at the same state. Our centered shaping strategy (§4) explicitly removes this offset by subtracting a state-dependent baseline, ensuring the shaping signal reflects only the relative action preferences.

### 3.5. Robustness to Estimation Error

Since score matching is approximate, we bound the impact of score estimation error $\eta$ on the recovered preferences.

**Theorem 3.11** (Lipschitz Continuity of Preferences). *Assume the learned score satisfies $\|\mathcal{S}_\phi(\boldsymbol{a}, \boldsymbol{s}) - \mathcal{S}^*(\boldsymbol{a}, \boldsymbol{s})\|_2 \leq \eta$ uniformly. Let $\Delta E(\boldsymbol{a}, \boldsymbol{a}') = E(\boldsymbol{a}, \boldsymbol{s}) - E(\boldsymbol{a}', \boldsymbol{s})$ be the relative preference between two actions at the same state. Then:*

$$
|\Delta E_\phi(\boldsymbol{a}, \boldsymbol{a}') - \Delta E^*(\boldsymbol{a}, \boldsymbol{a}')| \leq \eta \cdot \|\boldsymbol{a} - \boldsymbol{a}'\|_2. \tag{13}
$$

☐ *Proof in Appendix A.4.*

*Remark* 3.12 (On the Lipschitz Assumption). The uniform bound $\|\mathcal{S}_\phi(\boldsymbol{a}, \boldsymbol{s}) - \mathcal{S}^*(\boldsymbol{a}, \boldsymbol{s})\|_2 \leq \eta$ is mild and typically satisfied in practice. Neural networks with bounded weights and Lipschitz activation functions are inherently Lipschitz continuous (Gouk et al., 2021).

This result confirms that our method degrades gracefully. Small errors in the score field translate to bounded errors in action ranking, scaling linearly with the distance between actions. In the context of downstream RL, this means that for actions within a bounded action space of diameter $\text{diam}(\mathcal{A})$, the maximum reward estimation error per step is $\alpha \cdot \epsilon \cdot \text{diam}(\mathcal{A})$, which remains controlled as long as score matching is accurate.

## 4. Methodology

**Architecture** The theoretical constraints identified in Sec. 3 directly lead to our architectural choices. To satisfy the conservative field requirement (§3.2), we do not

**Algorithm 2** ENERGYFLOW Action Generation

**Require:** State $\boldsymbol{s}$, trained $E_\phi$, steps $K$, endpoint $\gamma = 10^{-3}$
1: Sample $\boldsymbol{a}_T \sim \mathcal{N}(\boldsymbol{0}, \sigma^2(T)\boldsymbol{I})$
2: $\Delta t \leftarrow (T - \gamma)/K$
3: **for** $k = 0, \dots, K-1$ **do**
4:    $t_k \leftarrow T - k\Delta t$
5:    $\boldsymbol{g}_k \leftarrow \frac{1}{2}\frac{d[\sigma^2(t_k)]}{dt}\nabla_{\boldsymbol{a}} E_\phi(\boldsymbol{a}_k, \boldsymbol{s}, t_k)$
6:    $\boldsymbol{a}_{k+1} \leftarrow \boldsymbol{a}_k - \Delta t \cdot \boldsymbol{g}_k$
7: **end for**
8: **return** Action $\boldsymbol{a}_K$, Energy $E_\phi(\boldsymbol{a}_K, \boldsymbol{s}, \gamma)$

directly regress the vector-valued score. Instead, we parameterize a scalar energy function $E_\phi : \mathcal{A} \times \mathcal{S} \times [0, T] \to \mathbb{R}$ and obtain the score via automatic differentiation:

$$\mathcal{S}_\phi(\boldsymbol{a}, \boldsymbol{s}, t) := -\nabla_{\boldsymbol{a}} E_\phi(\boldsymbol{a}, \boldsymbol{s}, t). \tag{14}$$

By construction, $\nabla_{\boldsymbol{a}} \times \mathcal{S}_\phi \equiv 0$, ensuring that learned preferences remain transitive and physically realizable. Detailed network implementation can be found in Appendix C.1.

**Training Paradigm** We estimate the energy landscape using denoising score matching. Following the variance-exploding formulation with noise schedule $\sigma(t) = \sigma_{\min}^{1-t/T}\sigma_{\max}^{t/T}$ (where $\sigma_{\min} = 0.01$, $\sigma_{\max} = 10.0$, $T = 1.0$), we minimize:

$$\mathcal{L}(\phi) = \mathbb{E}_{t, \boldsymbol{a}_0, \boldsymbol{\varepsilon}}\left[\sigma^2(t)\left\|-\nabla_{\boldsymbol{a}_t} E_\phi(\boldsymbol{a}_t, \boldsymbol{s}, t) + \frac{\boldsymbol{\varepsilon}}{\sigma(t)}\right\|^2\right], \tag{15}$$

where $\boldsymbol{a}_t = \boldsymbol{a}_0 + \sigma(t)\boldsymbol{\varepsilon}$, with $t \sim \mathcal{U}[0, T]$, $(\boldsymbol{s}, \boldsymbol{a}_0) \sim \mathcal{D}$, $\boldsymbol{\varepsilon} \sim \mathcal{N}(\boldsymbol{0}, \boldsymbol{I})$, and $\lambda(t) = \sigma^2(t)$ ensures uniform contribution across noise levels. As $t \to 0$, minimizing this objective is equivalent to recovering the maximum-entropy reward gradient (Theorem 3.3).

**Reward Extraction** While Proposition 3.9 states that the raw energy $E_\phi$ preserves within-state action rankings, the arbitrary offset $c(\boldsymbol{s})$ introduces high variance when $E_\phi$ is used as a reward signal in downstream RL. To mitigate this, we introduce *centered shaping*:

$$\tilde{r}_\phi(\boldsymbol{a}, \boldsymbol{s}) := -\left(E_\phi(\boldsymbol{a}, \boldsymbol{s}, \epsilon) - \underbrace{\mathbb{E}_{\boldsymbol{a}' \sim \mathcal{N}(\boldsymbol{0}, \boldsymbol{I})}[E_\phi(\boldsymbol{a}', \boldsymbol{s}, \gamma)]}_{\text{State-dependent Baseline}}\right), \tag{16}$$

where $\gamma = 10^{-3}$ is the ODE endpoint. By subtracting the expected energy under a reference distribution, we effectively normalize the state-dependent offset, centering the reward at every state. This ensures the shaping signal reflects only relative action preferences at a given state.

The baseline is approximated via Monte Carlo sampling with $M = 16$ samples from $\mathcal{N}(\boldsymbol{0}, \boldsymbol{I})$ (Actions are stan-

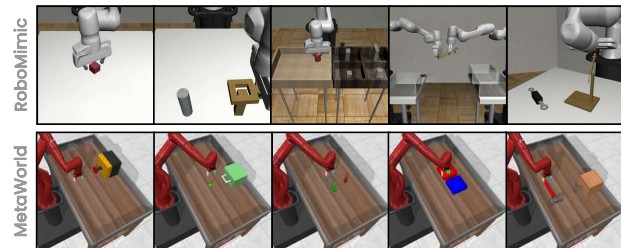

*Figure 2.* **Evaluation tasks.** We test on manipulation tasks spanning varying difficulty on RoboMimic and Meta-World.

dardized to approximately unit variance; see §5.1). Unlike methods that require stochastic trace estimation (e.g., Hutchinson's estimator for CNF log-likelihoods) (Grathwohl et al., 2019), our baseline computation is deterministic for a fixed set of reference samples, yielding a low-variance reward signal for policy gradient updates.

# 5. Experiments

We design our experimental evaluation to address following research questions: **RQ1:** Does explicit energy parameterization preserve strong behavior cloning performance? **RQ2:** Can the energy-parameterized policy transfer to real-world robotic manipulation tasks? **RQ3:** Can the learned energy serve as an effective reward signal for downstream reinforcement learning? **RQ4:** Does integrability improve robustness under distribution shift, as predicted by Lemma 3.8? **RQ5:** How sensitive is ENERGYFLOW to hyperparameters? **RQ6:** Does ENERGYFLOW achieve competitive inference speed compared to existing methods?

## 5.1. Experimental Setup

**Simulation Benchmarks.** We evaluate our approach on two widely used manipulation benchmarks RoboMimic (Mandlekar et al., 2021) and Meta-World (McLean et al., 2025). Specifically, we evaluate on five RoboMimic tasks (Lift, Can, Square, Transport, ToolHang) and five Meta-World tasks (ButtonPress, DrawerOpen, Assembly, BinPicking, Hammer). Figure 2 illustrates the complete task suite. These environments span a range of difficulty levels, from simple pick-and-place operations to complex multi-stage manipulation requiring precise coordination. Detailed task descriptions are provided in Appendix D.1. Following standard practice (Zhao et al., 2023; Chi et al., 2023), all actions are standardized to zero mean and unit variance using statistics computed from the training demonstrations.

**Baselines.** We compare ENERGYFLOW against a comprehensive set of baselines spanning three categories. We include **autoregressive policies**: LSTM-GMM (Dalal et al., 2023), which combines recurrent temporal modeling with Gaussian mixture outputs for multimodal action predic-

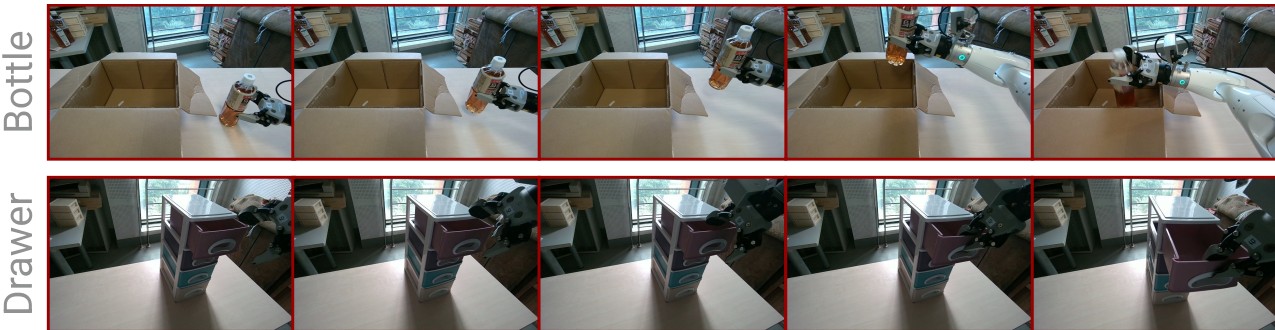

*Figure 3.* **Real-world evaluation tasks.** We evaluate on two contact-rich manipulation tasks: **Bottle** (top), where the robot must grasp a bottle and place it into a cardboard box, and **Drawer** (bottom), where the robot must pull the drawer open.

tion; **generative policies**: Diffusion Policy (Chi et al., 2023), which learns action distributions through iterative denoising, and Flow Policy (Zhang et al., 2025b), which employs continuous normalizing flows for density estimation; **energy-based methods**: Implicit BC (IBC) (Florence et al., 2021), which parameterizes policies implicitly through energy minimization and EBT-Policy (Davies et al., 2025), which combines energy-based modeling with transformer architectures; **inverse reinforcement learning methods**: EBIL (Liu et al., 2021), NEAR (Diwan et al., 2025), and IQ-Learn (Garg et al., 2021), which recover reward functions from demonstrations through different adversarial or information-theoretic objectives. The detailed implementation of these baselines are in C.5.

### 5.2. Imitation Learning Performance (RQ1)

Tables 1 and 2 report success rates on RoboMimic and Meta-World benchmarks respectively. On RoboMimic, EN-ERGYFLOW achieves the highest average success rate of 93.8%, outperforming Diffusion Policy (91.2%) and Flow Policy (89.6%). The improvements are particularly large on challenging tasks: ENERGYFLOW achieves 84.2% on Tool-Hang compared to 77.2% for Diffusion Policy. On Meta-World, ENERGYFLOW similarly leads with 92.5% average success, demonstrating consistent performance across diverse manipulation scenarios. Demonstrations of these tasks can be found in Appendix E.1. Notably, ENERGYFLOW also outperforms existing energy-based approaches. These results indicates that our conservative parameterization and flow-matching training objective can further enhance energy-based policy representation.

### 5.3. Real Robot Deployment (RQ2)

To validate real-world applicability, we deploy ENER-GYFLOW on a physical robot platform and evaluate whether the learned energy-parameterized policy can transfer effectively to contact-rich manipulation scenarios. Specifi-

*Table 1.* Success rates (%) on RoboMimic tasks (ph). Mean ± std over 3 seeds. **Bold**: best, underline: second best.

| Method | Lift | Can | Square | Transport | ToolHang | Avg. |
|---|---|---|---|---|---|---|
| LSTM-GMM | 97.8±1.7 | 71.4±8.4 | 64.3±2.3 | 65.6±4.9 | 46.0±6.0 | 69.0 |
| Diffusion Policy | **100.0**±0.0 | 99.2±0.2 | 93.5±0.6 | 85.9±1.5 | 77.2±1.2 | 91.2 |
| Flow Policy | 99.6±0.4 | 98.4±0.8 | 91.8±1.2 | 83.6±2.0 | 74.8±2.4 | 89.6 |
| EBT Policy | 96.2±1.6 | 88.6±3.2 | 78.4±3.8 | 72.4±4.2 | 58.6±4.4 | 78.8 |
| EBIL | 92.4±3.2 | 76.8±5.4 | 58.2±6.2 | 48.6±5.8 | 32.4±6.4 | 61.7 |
| NEAR | 93.6±2.8 | 78.4±4.8 | 71.4±5.6 | 52.2±5.4 | 36.8±5.8 | 66.5 |
| IQ-Learn | 95.2±2.2 | 82.6±4.2 | 68.8±4.8 | 58.4±4.6 | 44.2±5.2 | 69.8 |
| Implicit BC | 70.9±20.8 | 30.8±2.6 | 10.2±0.1 | 0.0±0.0 | 0.0±0.0 | 22.4 |
| Ours | **100.0**±0.0 | **100.0**±0.0 | **95.3**±0.5 | **89.4**±1.6 | **84.2**±1.4 | **93.8** |

*Table 2.* Success rates (%) on Meta-World tasks. Mean ± std over 5 seeds. **Bold**: best, underline: second best.

| Method | Button | Drawer | Assembly | Bin | Hammer | Avg. |
|---|---|---|---|---|---|---|
| LSTM-GMM | 80.2±4.2 | 74.6±4.6 | 48.4±5.8 | 66.8±5.2 | 70.6±4.8 | 68.1 |
| Diffusion Policy | **100.0**±0.0 | 93.6±1.6 | 76.4±3.4 | 89.6±2.2 | 94.0±1.8 | 90.7 |
| Flow Policy | **100.0**±0.0 | 92.8±1.8 | 74.8±3.6 | 87.6±2.4 | 92.2±2.0 | 89.5 |
| EBT-Policy | 84.2±3.4 | 81.6±3.8 | 62.4±4.8 | 75.8±4.2 | 85.0±3.6 | 77.8 |
| EBIL | 74.6±5.4 | 68.2±5.8 | 38.6±6.6 | 58.4±6.0 | 64.8±5.6 | 60.9 |
| NEAR | 76.8±5.0 | 70.4±5.4 | 42.2±6.2 | 61.6±5.6 | 67.0±5.2 | 63.6 |
| IQ-Learn | 76.4±4.2 | 72.8±4.6 | 52.6±5.4 | 66.2±5.0 | 76.5±4.4 | 68.9 |
| Implicit BC | 28.4±8.2 | 24.6±7.4 | 12.8±5.6 | 18.2±6.8 | 26.0±7.8 | 22.0 |
| Ours | **100.0**±0.0 | **94.2**±1.4 | **82.6**±2.8 | **90.9**±1.9 | **94.6**±1.5 | **92.5** |

cally, we conduct experiments using AGIBOT G1 robot [1] equipped with 7-DoF arms and parallel-jaw gripper. Visual observations are captured by a single RGB camera mounted fixed at head. We evaluate on two manipulation tasks Bottle and Drawer with 20 expert demonstration trajectories. Our ENERGYFLOW obtained 100% success rate on both tasks, with 3 initial position change, each with 20 rollouts. One success trajectory of ENERGYFLOW for each task is shown in Figure 3. Qualitatively, we observe that ENERGYFLOW produces smoother trajectories with fewer hesitations near contact points. More details about the real robot experiment are in Appendix E.2.

### 5.4. Reward Quality (RQ3)

A central advantage of our framework is that the learned energy function serves as reward signal for reinforcement learning, enabling policy training without access to ground-

---

[1] https://www.agibot.com/products/G1

truth environment rewards. We evaluate this by training Soft Actor-Critic (SAC) (Haarnoja et al., 2018) agents for 200k environment steps on RoboMimic Square and Transport. Detailed protocols are provided in Appendix C.6.

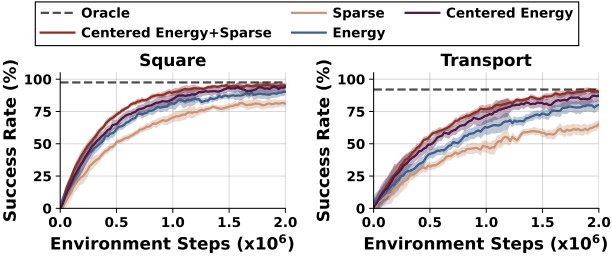

*Figure 4.* **SAC training using different reward signals.** We compare our energy-based rewards against sparse task signals and oracle dense rewards.

Figure 4 compares our centered shaping with sparse task rewards, raw energy rewards, and oracle dense rewards. With sparse rewards, the agent gets no signal until it succeeds by chance, which makes early training slow and noisy. Raw energy rewards are dense, but they do not reliably push the agent toward the goal: maximizing likelihood under demonstrations can encourage staying in common states instead of making progress, leading to early plateaus. Our centered formulation (Eq. 16) fixes this by basing reward on state transitions rather than state density, so the learned energy directly encourages forward progress and achieves near-oracle success on both tasks. Notably, *Centered Energy+Sparse* performs best, suggesting that the energy reward provides step-by-step guidance, while the sparse reward ensures the policy still optimizes for task completion.

### 5.5. Out-of-Distribution Generalization (RQ4)

To validate Lemma 3.8, which posits that conservative fields generalize better to novel states, we evaluate performance under increasing initial position perturbations (levels 0, S, M, L; see Appendix C.7). As shown in Figure 5, while all methods can achieve high performance in-distribution, EN-ERGYFLOW demonstrates superior stability as perturbation magnitude increases. Across these tasks, ENERGYFLOW outperforms Diffusion and Flow Policy baselines at medium and large perturbation levels, maintaining robust success rates where unconstrained models degrade. These results confirm that the curl-free constraint acts as a powerful geometric regularizer, preventing the learning of latent artifacts and improving extrapolation in tasks with spatial variability.

### 5.6. Reward Extraction Sensitivity (RQ5)

Table 3 analyzes sensitivity to the time parameter $\gamma$ used for energy evaluation. Performance remains robust across three orders of magnitude ($\gamma \in [10^{-4}, 10^{-2}]$). Degradation

*Table 3.* Sensitivity to reward extraction time $\gamma$. Success rate (%) on RoboMimic Square after 200K SAC steps.

| $\gamma$ | $10^{-4}$ | $10^{-3}$ | $10^{-2}$ | $10^{-1}$ | 0.5 |
|---|---|---|---|---|---|
| Success (%) | $94.2_{\pm 2.6}$ | $\mathbf{95.3_{\pm 2.4}}$ | $88.4_{\pm 1.8}$ | $78.4_{\pm 3.4}$ | $72.6_{\pm 4.8}$ |

*Table 4.* Inference comparison on RoboMimic Square. Latency measured for 10Hz control on NVIDIA A100.

| Method | Success (%) ↑ | Latency (ms) ↓ | Exposes Scalar |
|---|---|---|---|
| Implicit BC (50 Langevin) | $10.2_{\pm 3.2}$ | 52.4 | ✔ |
| Implicit BC (10 Langevin) | $0.0_{\pm 0.0}$ | 12.8 | ✔ |
| Diffusion Policy (100 DDPM) | $93.5_{\pm 3.2}$ | 32.4 | ✗ |
| Diffusion Policy (20 DDIM) | $90.4_{\pm 4.6}$ | 9.1 | ✗ |
| Flow Policy | $91.8_{\pm 1.4}$ | 8.2 | ✗ |
| ENERGYFLOW ($K = 10$) | $94.0_{\pm 1.8}$ | 9.8 | ✔ |
| ENERGYFLOW ($K = 20$) | $\mathbf{95.3_{\pm 0.8}}$ | 11.4 | ✔ |

occurs only at larger values ($\gamma \geq 0.1$) where the noised distribution diverges significantly from the data distribution, thereby weakening the approximation of the score function.

### 5.7. Inference Efficiency (RQ6)

Table 4 demonstrates that ENERGYFLOW achieves a favorable balance of speed and utility. unlike Implicit BC, which requires computationally expensive Langevin sampling for high performance, ENERGYFLOW attains superior success rates with latency comparable to the non-energy-based Flow Policy. This confirms that ENERGYFLOW provides the benefits of an explicit energy function without the runtime prohibitive costs typically associated with EBMs.

## 6. Related Work

### 6.1. Generative Models for Behavior Cloning

Behavior cloning learns policies by directly mimicking expert demonstrations, with recent advances leveraging expressive generative models to capture multi-modal action distributions (Schaal, 1996; Wolf et al., 2025; Urain et al., 2026). Diffusion Policy (Chi et al., 2023) demonstrated that diffusion-based action generation significantly outperforms prior methods on contact-rich manipulation. Subsequent works have extended this framework to 3D visual manipulation (Ze et al., 2024), hierarchical planning (Chen et al., 2024), and language-conditioned policies (Wen et al., 2025). To address computational overhead, efficient variants based on consistency distillation (Prasad et al., 2024) and flow matching (Zhang et al., 2025a; Braun et al., 2024; Funk et al., 2024) have been proposed. While these policies excel at modeling complex distributions, they remain limited to imitating trajectories without capturing underlying intent, limiting generalization to out-of-distribution states (Ada et al., 2024; Zare et al., 2024). Our work adds an integrability constraint via explicit energy parameterization, complementing these efficiency-focused approaches while enabling reward extraction.

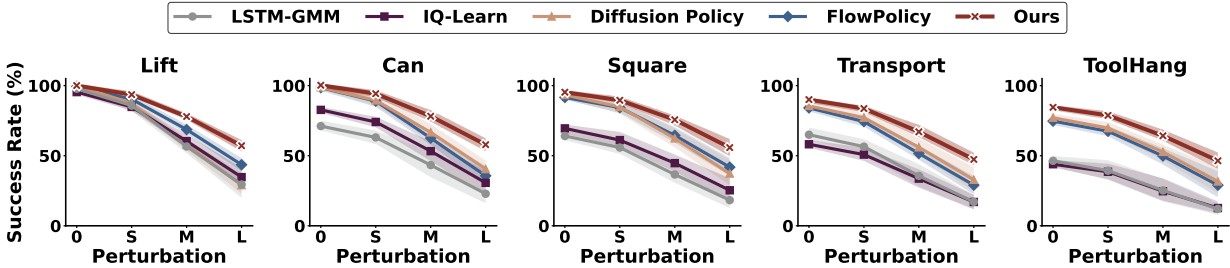

*Figure 5.* **OOD generalization on RoboMimic.** Success rate vs. initial position perturbation magnitude. ENERGYFLOW degrades more gracefully than baselines, with the gap widening at larger perturbations. Shaded regions indicate 95% confidence intervals.

## 6.2. Inverse Reinforcement Learning

Unlike behavior cloning, IRL seeks to recover the latent reward behind expert behavior. Classical methods such as maximum-entropy IRL (Ziebart et al., 2008) and Bayesian formulations (Ramachandran & Amir, 2007) suffered from computational intractability due to repeated policy optimization. Adversarial methods address this by casting reward learning as occupancy measure matching: GAIL (Ho & Ermon, 2016) and AIRL (Fu et al., 2018) enable scalable IRL through adversarial updates but inherit training instability and mode collapse issues (Wang et al., 2017). Energy-Based Models offer an alternative, directly parameterizing reward as a scalar energy (Song & Kingma, 2021; Du & Mordatch, 2019). While EBMs avoid adversarial dynamics, they require approximating intractable partition functions via expensive MCMC sampling, which scales poorly to high-dimensional action spaces. Recent works bridge generative modeling and IRL by replacing adversarial discriminators with diffusion models (Wang et al., 2024; Lai et al., 2024; Wan et al., 2025). However, these approaches treat diffusion as a drop-in discriminator replacement rather than exploiting the deeper connection between denoising and energy landscapes.

## 6.3. Energy-Based Imitation Learning

Energy-based formulations perform expert imitation as learning a scalar function whose minima correspond to expert-like actions or trajectories. In this view, energy can play two distinct roles in imitation learning: (i) an implicit *policy parameterization* used directly for action selection, or (ii) a learned *surrogate reward* that is subsequently optimized by RL (Li et al., 2025).

On the policy side, Implicit Behavioral Cloning (IBC) learns an energy over state-action pairs and predicts actions by minimizing this energy without explicit likelihood modeling (Florence et al., 2021). Subsequent work improves training stability through contrastive objectives and refined negative-sampling schemes (Singh et al., 2024; Antonelo et al., 2025), while EBT-Policy scales this paradigm with transformer-based energy functions and iterative inference,

achieving strong robustness with fewer inference steps than diffusion policies (Davies et al., 2025). However, these methods treat the learned energy as a *decision score* rather than an *identifiable reward*, and inference relies on iterative optimization whose dynamics need not correspond to a well-defined potential.

On the reward-learning side, maximum-entropy inverse optimal control can be interpreted through an energy perspective, where costs define an unnormalized trajectory distribution (Ziebart et al., 2008; Fu et al., 2018; Finn et al., 2016). EBIL (Liu et al., 2021) makes this connection explicit and proposes a two-stage pipeline: first estimate the expert energy via score matching, then treats the recovered energy as a reward for downstream maximum-entropy RL. Related approaches such as NEAR (Diwan et al., 2025) similarly learn energy-based rewards and then perform policy optimization. While these methods can yield explainable reward signals, they do not leverage the deeper structural link between denoising dynamics and conservative energy landscapes for *simultaneous* policy learning and reward recovery.

## 7. Conclusion

We propose ENERGYFLOW, a framework that bridges diffusion-based imitation learning and inverse reinforcement learning through energy-based parameterization. Our theoretical analysis establishes three key results: (1) the score function of an optimal policy encodes the gradient of its soft Q-function, enabling reward recovery via score matching without adversarial optimization; (2) constraining the learned vector field to be conservative, provably reduces hypothesis complexity and improves generalization; and (3) score estimation errors translate to bounded errors in action preferences, avoiding degradation under approximate learning. Our empirical findings validate these theoretical insights. ENERGYFLOW matches or exceeds state-of-the-art diffusion policies on standard benchmarks while simultaneously exposing a scalar energy that serves as an effective reward signal for policy refinement. Notably, the conservative constraint yields substantial out-of-distribution robustness without sacrificing in-distribution performance.

## Impact Statement

This paper presents work whose goal is to advance the field of machine learning. There are many potential societal consequences of our work, none of which we feel must be specifically highlighted here.

## Acknowlegments

This paper is supported by Fundamental Research Funds for the Central Universities (No.YG2024ZD06), NSFC (No.62176155) and Shanghai Municipal Science and Technology Major Project, China, under grant No. 2021SHZDZX0102. D. J. acknowledges the support from the ZJU Kunpeng & Ascend Center of Excellence, and the Dream Set Off - Kunpeng & Ascend Seed Program.

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

# A. Proofs

## A.1. Proof of Theorem 3.6

**Theorem 3.6 (Complexity Reduction via Conservative Constraints).** *Let $\phi : \mathbb{R}^{in} \to \mathbb{R}^k$ be a neural feature representation with bounded feature norm $\sup_{\boldsymbol{x}} \|\phi(\boldsymbol{x})\|_2 \leq B$ and bounded Jacobian Frobenius norm $\sup_{\boldsymbol{x}} \|J_\phi(\boldsymbol{x})\|_F \leq L$. Let $\mathcal{F}_{unc}$ be the class of arbitrary linear vector fields over $\phi$, and $\mathcal{F}_{cons}$ be the class of conservative vector fields (gradients of potentials over $\phi$). The Empirical Rademacher complexity of the conservative class is strictly tighter with respect to the output dimension $d$:*

$$\hat{\mathfrak{R}}_S(\mathcal{F}_{\text{unc}}) \leq \frac{\Lambda B \sqrt{d}}{\sqrt{n}}, \tag{17}$$

$$\hat{\mathfrak{R}}_S(\mathcal{F}_{\text{cons}}) \leq \frac{\Lambda L}{\sqrt{n}}. \tag{18}$$

*For high-dimensional action spaces where $d$ is large, provided the representation is smooth ($L \ll B\sqrt{d}$), we have $\hat{\mathfrak{R}}_S(\mathcal{F}_{cons}) \ll \hat{\mathfrak{R}}_S(\mathcal{F}_{unc})$.*

*Proof.* Let $\mathcal{D} = \{\boldsymbol{x}_1, \ldots, \boldsymbol{x}_n\}$ be the dataset. The empirical Rademacher complexity is given by:

$$\hat{\mathfrak{R}}_{\mathcal{D}}(\mathcal{F}) = \frac{1}{n} \mathbb{E}_{\boldsymbol{\sigma}} \left[ \sup_{f \in \mathcal{F}} \sum_{i=1}^{n} \langle \boldsymbol{\sigma}_i, f(\boldsymbol{x}_i) \rangle \right], \tag{19}$$

where $\boldsymbol{\sigma}_i$ are independent Rademacher vectors in $\mathbb{R}^d$ such that $\mathbb{E}[\boldsymbol{\sigma}_i] = \mathbf{0}$ and $\|\boldsymbol{\sigma}_i\|^2 = d$.

**Analysis of Unconstrained Fields.** The unconstrained class consists of functions $f(\boldsymbol{x}) = \boldsymbol{W}\phi(\boldsymbol{x})$ where $\boldsymbol{W} \in \mathbb{R}^{d \times k}$ and $\|\boldsymbol{W}\|_F \leq \Lambda$.

$$n\hat{\mathfrak{R}}_{\mathcal{D}}(\mathcal{F}_{\text{unc}}) = \mathbb{E}_{\boldsymbol{\sigma}} \left[ \sup_{\|\boldsymbol{W}\|_F \leq \Lambda} \sum_{i=1}^{n} \langle \boldsymbol{\sigma}_i, \boldsymbol{W}\phi(\boldsymbol{x}_i) \rangle \right]$$

$$= \mathbb{E}_{\boldsymbol{\sigma}} \left[ \sup_{\|\boldsymbol{W}\|_F \leq \Lambda} \left\langle \boldsymbol{W}, \sum_{i=1}^{n} \boldsymbol{\sigma}_i \phi(\boldsymbol{x}_i)^\top \right\rangle_F \right].$$

By the Cauchy-Schwarz inequality for the Frobenius inner product, the supremum is attained when $\boldsymbol{W}$ is aligned with the random sum. Thus:

$$n\hat{\mathfrak{R}}_{\mathcal{D}}(\mathcal{F}_{\text{unc}}) \leq \Lambda \cdot \mathbb{E}_{\boldsymbol{\sigma}} \left[ \left\| \sum_{i=1}^{n} \boldsymbol{\sigma}_i \phi(\boldsymbol{x}_i)^\top \right\|_F \right].$$

Using Jensen's inequality and noting that cross-terms $\mathbb{E}[\langle \boldsymbol{\sigma}_i, \boldsymbol{\sigma}_j \rangle] = 0$ for $i \neq j$ vanish due to independence:

$$\mathbb{E} \left\| \sum_{i=1}^{n} \boldsymbol{\sigma}_i \phi(\boldsymbol{x}_i)^\top \right\|_F \leq \sqrt{\sum_{i=1}^{n} \mathbb{E}_{\boldsymbol{\sigma}} \left[ \|\boldsymbol{\sigma}_i\|^2 \|\phi(\boldsymbol{x}_i)\|^2 \right]}$$

$$= \sqrt{\sum_{i=1}^{n} d \cdot \|\phi(\boldsymbol{x}_i)\|^2}$$

$$\leq \sqrt{n \cdot d \cdot B^2} = B\sqrt{nd}.$$

Substituting this back yields the unconstrained bound:

$$\hat{\mathfrak{R}}_S(\mathcal{F}_{\text{unc}}) \leq \frac{\Lambda B \sqrt{d}}{\sqrt{n}}. \tag{20}$$

**Analysis of Conservative Fields.** The conservative class consists of functions $f(\boldsymbol{x}) = J_\phi(\boldsymbol{x})^\top \boldsymbol{w}$ (gradients of $E(\boldsymbol{x}) = \boldsymbol{w}^\top \phi(\boldsymbol{x})$), where $\boldsymbol{w} \in \mathbb{R}^k$ and $\|\boldsymbol{w}\|_2 \leq \Lambda$.

$$n\hat{\mathfrak{R}}_S(\mathcal{F}_{\text{cons}}) = \mathbb{E}_{\boldsymbol{\sigma}}\left[\sup_{\|\boldsymbol{w}\|_2 \leq \Lambda} \sum_{i=1}^n \langle \boldsymbol{\sigma}_i, J_\phi(\boldsymbol{x}_i)^\top \boldsymbol{w}\rangle\right]$$

$$= \mathbb{E}_{\boldsymbol{\sigma}}\left[\sup_{\|\boldsymbol{w}\|_2 \leq \Lambda} \left\langle \boldsymbol{w}, \sum_{i=1}^n J_\phi(\boldsymbol{x}_i)\boldsymbol{\sigma}_i\right\rangle\right].$$

By Cauchy-Schwarz inequality in Euclidean space:

$$n\hat{\mathfrak{R}}_S(\mathcal{F}_{\text{cons}}) \leq \Lambda \cdot \mathbb{E}_{\boldsymbol{\sigma}}\left[\left\|\sum_{i=1}^n J_\phi(\boldsymbol{x}_i)\boldsymbol{\sigma}_i\right\|_2\right].$$

Again applying Jensen's inequality and independence of $\boldsymbol{\sigma}_i$:

$$\mathbb{E}\left\|\sum_{i=1}^n J_\phi(\boldsymbol{x}_i)\boldsymbol{\sigma}_i\right\|_2 \leq \sqrt{\sum_{i=1}^n \mathbb{E}_{\boldsymbol{\sigma}}\left[\boldsymbol{\sigma}_i^\top J_\phi(\boldsymbol{x}_i)^\top J_\phi(\boldsymbol{x}_i)\boldsymbol{\sigma}_i\right]}.$$

Using the property that for Rademacher vectors $\mathbb{E}[\boldsymbol{\sigma}^\top \boldsymbol{A}\boldsymbol{\sigma}] = \text{Tr}(\boldsymbol{A})$, we have:

$$\mathbb{E}[\boldsymbol{\sigma}_i^\top J_\phi(\boldsymbol{x}_i)^\top J_\phi(\boldsymbol{x}_i)\boldsymbol{\sigma}_i] = \text{Tr}(J_\phi(\boldsymbol{x}_i)^\top J_\phi(\boldsymbol{x}_i))$$

$$= \|J_\phi(\boldsymbol{x}_i)\|_F^2 \leq L^2.$$

Thus:

$$n\hat{\mathfrak{R}}_S(\mathcal{F}_{\text{cons}}) \leq \Lambda\sqrt{nL^2} = \Lambda L\sqrt{n}.$$

Yielding the conservative bound:

$$\hat{\mathfrak{R}}_S(\mathcal{F}_{\text{cons}}) \leq \frac{\Lambda L}{\sqrt{n}}. \tag{21}$$

Comparing Eq. (20) and Eq. (21), the unconstrained complexity scales explicitly with $\sqrt{d}$ (the square root of the output dimension). In contrast, the conservative complexity scales with $L$ (the smoothness of the representation).

Since neural network representations generally learn smooth manifolds where the tangent space volume (captured by $L$) grows significantly slower than the ambient dimension $d$, the conservative constraint provides a structurally superior generalization guarantee. $\square$

### A.2. Proof of Lemma 3.8

**Lemma 3.8 (OOD Generalization).** *Let $\mathcal{D}_S$ be the source training distribution and $\mathcal{D}_T$ be a target (OOD) distribution. Let $h^* \in \mathcal{F}_{cons}$ be the ground truth conservative field. Assume that all hypotheses in $\mathcal{F}_{cons}$ and $\mathcal{F}_{unc}$ are uniformly bounded by $M > 0$. For any learned hypothesis $f$, let the risk be $\mathcal{R}_\mathcal{D}(f) = \mathbb{E}_{\boldsymbol{x}\sim\mathcal{D}}[\|f(\boldsymbol{x}) - h^*(\boldsymbol{x})\|^2]$. The risk on the target domain for the conservative estimator satisfies, with probability at least $1 - \delta$:*

$$\mathcal{R}_{\mathcal{D}_T}(\hat{f}_{\text{cons}}) \leq \hat{\mathcal{R}}_S(\hat{f}_{\text{cons}}) + \frac{1}{2}d_{\mathcal{H}\Delta\mathcal{H}}(\mathcal{D}_S, \mathcal{D}_T) + \mathcal{O}\left(\frac{M\Lambda L}{\sqrt{n}}\right),$$

*whereas for the unconstrained estimator $\hat{f}_{unc}$, the complexity term scales with $\mathcal{O}(M\Lambda B\sqrt{d}/\sqrt{n})$.*

*Proof.* The proof relies on combining the standard generalization bounds based on Rademacher complexity with the domain adaptation theory introduced by Ben-David et al. (2010).

For any hypothesis $h$ in a hypothesis class $\mathcal{H}$, the relationship between the risk on the target distribution $\mathcal{R}_{\mathcal{D}_T}(h)$ and the source distribution $\mathcal{R}_{\mathcal{D}_S}(h)$ is bounded by the discrepancy between the domains. Specifically:

$$\mathcal{R}_{\mathcal{D}_T}(h) \leq \mathcal{R}_{\mathcal{D}_S}(h) + \frac{1}{2}d_{\mathcal{H}\Delta\mathcal{H}}(\mathcal{D}_S, \mathcal{D}_T) + \lambda, \tag{22}$$

where $d_{\mathcal{H}\Delta\mathcal{H}}$ is the discrepancy distance and $\lambda$ is the combined error of the ideal joint hypothesis. Since we assume the ground truth $h^*$ belongs to the conservative class $\mathcal{F}_{\mathrm{cons}}$, the ideal error $\lambda$ is negligible for the conservative estimator.

Eq. (22) relates the *true* population risks. However, learning algorithms minimize the *empirical* source risk $\hat{\mathcal{R}}_S(h)$ on a dataset of size $n$. Standard learning theory bounds the true source risk as:

$$\mathcal{R}_{\mathcal{D}_S}(h) \leq \hat{\mathcal{R}}_S(h) + 2\mathfrak{R}_S(\ell \circ \mathcal{H}) + \sqrt{\frac{\log(1/\delta)}{2n}}, \tag{23}$$

where $\mathfrak{R}_S(\ell \circ \mathcal{H})$ is the Rademacher complexity of the loss composed with the hypothesis class.

The risk is defined using the squared $L_2$ loss: $\ell(f(\boldsymbol{x}), h^*(\boldsymbol{x})) = \|f(\boldsymbol{x}) - h^*(\boldsymbol{x})\|_2^2$. The squared loss is not globally Lipschitz, but under the boundedness assumption ($\|f(\boldsymbol{x})\|_2 \leq M$ and $\|h^*(\boldsymbol{x})\|_2 \leq M$ for all $\boldsymbol{x}$), the loss is restricted to a bounded domain where:

$$|\ell(y_1) - \ell(y_2)| = |y_1^2 - y_2^2| = |y_1 + y_2||y_1 - y_2| \leq 2M|y_1 - y_2|.$$

Thus, on the bounded domain, the squared loss is Lipschitz with constant $2M$. By Talagrand's contraction lemma:

$$\mathfrak{R}_S(\ell \circ \mathcal{H}) \leq 2M \cdot \mathfrak{R}_S(\mathcal{H}).$$

We now substitute the specific Rademacher complexity bounds derived in Theorem 3.6.

*Case A: Unconstrained Vector Fields ($\mathcal{F}_{unc}$).* Theorem 3.6 gives $\hat{\mathfrak{R}}_S(\mathcal{F}_{\mathrm{unc}}) \leq \frac{\Lambda B\sqrt{d}}{\sqrt{n}}$. Substituting into Eq. (23):

$$\mathrm{GenGap}(\hat{f}_{\mathrm{unc}}) \in \mathcal{O}\left(\frac{M\Lambda B\sqrt{d}}{\sqrt{n}}\right). \tag{24}$$

*Case B: Conservative Vector Fields ($\mathcal{F}_{cons}$).* Theorem 3.6 gives the tighter bound $\hat{\mathfrak{R}}_S(\mathcal{F}_{\mathrm{cons}}) \leq \frac{\Lambda L}{\sqrt{n}}$. Substituting into Eq. (23):

$$\mathrm{GenGap}(\hat{f}_{\mathrm{cons}}) \in \mathcal{O}\left(\frac{M\Lambda L}{\sqrt{n}}\right). \tag{25}$$

Combining the domain adaptation bound (Eq. (22)) with the complexity-based generalization gap, for the conservative estimator:

$$\mathcal{R}_{\mathcal{D}_T}(\hat{f}_{\mathrm{cons}}) \leq \hat{\mathcal{R}}_S(\hat{f}_{\mathrm{cons}}) + \mathrm{GenGap}(\hat{f}_{\mathrm{cons}}) + \frac{1}{2}d_{\mathcal{H}\Delta\mathcal{H}}(\mathcal{D}_S, \mathcal{D}_T) \tag{26}$$

$$= \hat{\mathcal{R}}_S(\hat{f}_{\mathrm{cons}}) + \frac{1}{2}d_{\mathcal{H}\Delta\mathcal{H}}(\mathcal{D}_S, \mathcal{D}_T) + \mathcal{O}\left(\frac{M\Lambda L}{\sqrt{n}}\right). \tag{27}$$

For the unconstrained estimator, the complexity term scales with $\sqrt{d}$. Thus, as $d \to \infty$, the bound for the unconstrained field diverges, while the conservative bound remains controlled by the smoothness parameter $L$. $\qquad\square$

### A.3. Proof of Proposition 3.9

**Proposition 3.9 (Within-State Action Ranking).** *The learned energy provides exact within-state action rankings:*

1. *Within-state ranking is exact: For any fixed state $\boldsymbol{s}$, $\arg\min_{\boldsymbol{a}} E_\phi(\boldsymbol{a}, \boldsymbol{s}) = \arg\max_{\boldsymbol{a}} Q^*(\boldsymbol{s}, \boldsymbol{a})$.*

2. *Cross-state comparison is ambiguous: The difference $E_\phi(\boldsymbol{a}, \boldsymbol{s}) - E_\phi(\boldsymbol{a}', \boldsymbol{s}')$ includes the unknown quantity $c(\boldsymbol{s}) - c(\boldsymbol{s}')$.*

*Proof.* From Theorem 3.3, the learned energy satisfies:

$$E_\phi(\boldsymbol{a}, \boldsymbol{s}) = -\frac{Q^*(\boldsymbol{s}, \boldsymbol{a})}{\alpha} + c(\boldsymbol{s}), \tag{28}$$

where $c(\boldsymbol{s})$ is a state-dependent constant arising from integration.

**Within-state ranking.** For a fixed state $s$, consider any two actions $a_1, a_2 \in A$. The energy difference is:

$$E_\phi(a_1, s) - E_\phi(a_2, s) = \left(-\frac{Q^*(s, a_1)}{\alpha} + c(s)\right) - \left(-\frac{Q^*(s, a_2)}{\alpha} + c(s)\right) \tag{29}$$

$$= -\frac{1}{\alpha}\left(Q^*(s, a_1) - Q^*(s, a_2)\right). \tag{30}$$

The state-dependent constant $c(s)$ cancels. Since $\alpha > 0$:

$$E_\phi(a_1, s) < E_\phi(a_2, s) \iff Q^*(s, a_1) > Q^*(s, a_2).$$

Therefore, $\arg\min_a E_\phi(a, s) = \arg\max_a Q^*(s, a)$.

**Cross-state ambiguity.** For two different states $s \neq s'$ and actions $a, a'$:

$$E_\phi(a, s) - E_\phi(a', s') = -\frac{Q^*(s, a)}{\alpha} + c(s) + \frac{Q^*(s', a')}{\alpha} - c(s') \tag{31}$$

$$= -\frac{1}{\alpha}\left(Q^*(s, a) - Q^*(s', a')\right) + \underbrace{(c(s) - c(s'))}_{\text{unknown}}. \tag{32}$$

The term $c(s) - c(s')$ cannot be determined from observations of expert behavior, as demonstrations reveal only which actions are preferred at each state, not the relative value of different states. $\square$

### A.4. Proof of Theorem 3.11

**Theorem 3.11 (Lipschitz Continuity of Preferences).** *Assume the learned score satisfies $\|S_\phi(a, s) - S^*(a, s)\|_2 \leq \epsilon$ uniformly. Let $\Delta E(a, a') = E(a, s) - E(a', s)$ be the relative preference between two actions at the same state. Then:*

$$|\Delta E_\phi(a, a') - \Delta E^*(a, a')| \leq \epsilon \cdot \|a - a'\|_2.$$

*Proof.* For an energy-based model with $p(a|s) \propto \exp(-E(a, s))$, the score function is:

$$S^*(a, s) = \nabla_a \log p(a|s) = -\nabla_a E^*(a, s). \tag{33}$$

Similarly, for the learned model: $\nabla_a E_\phi(a, s) = -S_\phi(a, s)$.

Define the quantity of interest:

$$\delta = |\Delta E_\phi(a, a') - \Delta E^*(a, a')|. \tag{34}$$

By the fundamental theorem of calculus for line integrals, the difference in a scalar potential between two points equals the line integral of its gradient. Let $\gamma(t) = a' + t(a - a')$ for $t \in [0, 1]$ be the straight-line path from $a'$ to $a$. Then:

$$E(a, s) - E(a', s) = \int_0^1 \nabla_a E(\gamma(t), s) \cdot (a - a') \, dt. \tag{35}$$

Substituting $\nabla_a E = -S$:

$$E(a, s) - E(a', s) = \int_0^1 -S(\gamma(t), s) \cdot (a - a') \, dt. \tag{36}$$

Therefore:

$$\delta = \left|\int_0^1 \left(S^*(\gamma(t), s) - S_\phi(\gamma(t), s)\right) \cdot (a - a') \, dt\right| \tag{37}$$

$$\leq \int_0^1 \left|\left(S^*(\gamma(t), s) - S_\phi(\gamma(t), s)\right) \cdot (a - a')\right| \, dt. \tag{38}$$

Applying the Cauchy-Schwarz inequality:

$$\delta \leq \int_0^1 \|\mathcal{S}^*(\gamma(t), \boldsymbol{s}) - \mathcal{S}_\phi(\gamma(t), \boldsymbol{s})\|_2 \cdot \|\boldsymbol{a} - \boldsymbol{a}'\|_2 \, dt. \tag{39}$$

Using the uniform error bound $\|\mathcal{S}_\phi - \mathcal{S}^*\|_2 \leq \epsilon$:

$$\delta \leq \int_0^1 \epsilon \cdot \|\boldsymbol{a} - \boldsymbol{a}'\|_2 \, dt \tag{40}$$

$$= \epsilon \cdot \|\boldsymbol{a} - \boldsymbol{a}'\|_2. \tag{41}$$

Thus, $|\Delta E_\phi(\boldsymbol{a}, \boldsymbol{a}') - \Delta E^*(\boldsymbol{a}, \boldsymbol{a}')| \leq \epsilon \cdot \|\boldsymbol{a} - \boldsymbol{a}'\|_2$. $\qquad\qquad\square$

## B. Baselines

To rigorously evaluate the efficacy of ENERGYFLOW, we compare against a diverse suite of baselines categorized by their underlying modeling paradigm. These methods represent the current state-of-the-art in imitation learning (IL) and inverse reinforcement learning (IRL):

**Explicit Autoregressive Policies.** We include **LSTM-GMM** (Dalal et al., 2023), a classic baseline that couples a Long Short-Term Memory (LSTM) network with a Gaussian Mixture Model (GMM) output head. This method explicitly maximizes the log-likelihood of expert actions. It serves as a benchmark for recurrent architectures that handle temporal dependencies but are constrained by the parametric assumptions of GMMs when modeling highly discontinuous action manifolds.

**Generative Policies.** To assess performance against modern generative modeling techniques, we compare against:

- **Diffusion Policy (DP)** (Chi et al., 2023): A state-of-the-art behavior cloning method that represents the policy as a conditional denoising diffusion probabilistic model. DP learns the gradient of the data distribution (score function) to iteratively denoise random noise into expert actions, offering superior stability and multimodal coverage compared to GANs.

- **Flow Policy** (Zhang et al., 2025b): A method utilizing continuous normalizing flows to learn complex action distributions via a sequence of invertible transformations. This baseline provides exact likelihood estimation and serves as a representative for bijective generative models.

**Energy-Based Models (EBMs).** We benchmark against methods that parameterize the policy implicitly via an energy function $E(\boldsymbol{s}, \boldsymbol{a})$:

- **Implicit BC (IBC)** (Florence et al., 2021): A non-parametric approach that learns an energy landscape where expert actions correspond to energy minima. IBC is particularly effective at capturing sharp discontinuities in the action space but relies on inference-time optimization (e.g., Langevin dynamics or CEM).

- **EBT-Policy** (Davies et al., 2025): An extension of EBMs that incorporates Transformer architectures. This baseline tests the importance of attention mechanisms in energy-based formulations for capturing long-horizon temporal dependencies.

**Inverse Reinforcement Learning (IRL).** Finally, we compare against methods that infer a reward function from demonstrations rather than cloning actions directly. We select **EBIL** (Liu et al., 2021), **NEAR** (Diwan et al., 2025), and **IQ-Learn** (Garg et al., 2021). These methods circumvent the instability of traditional adversarial training (e.g., GAIL) by deriving non-adversarial objectives. Specifically, IQ-Learn leverages the relationship between soft Q-learning and policy updates to recover rewards without a minimax game, serving as a strong baseline for sample-efficient reward recovery.

## C. Additional Implementation Details

### C.1. ENERGYFLOW Implementation

In this section, we detail the network architecture, training hyperparameters, and inference procedures for ENERGYFLOW. Our implementation relies on the PyTorch (Paszke et al., 2019) framework.

## C.2. Network Architecture

We adapt the 1D Conditional U-Net backbone from Diffusion Policy (Chi et al., 2023) to serve as our energy function parameterization. Unlike standard diffusion policies that directly regress the score (noise) field, our network approximates the scalar energy field $E_\phi(\boldsymbol{a}, \boldsymbol{s}, t)$, from which the score is derived via gradients.

**State and Time Encoding.** Since our setting involves low-dimensional state inputs (e.g., joint angles, velocities, object poses) without visual observations:

- **State Conditioning:** The observation sequence $\boldsymbol{s} \in \mathbb{R}^{T_{obs} \times D_s}$ is flattened and projected via a 2-layer MLP (Hidden dim: 128, Activation: Mish) into a conditioning vector $\boldsymbol{c}_{state}$.

- **Time Embedding:** The diffusion timestep $t$ is encoded using sinusoidal positional embeddings followed by a linear projection to match the channel dimensions of the U-Net blocks.

**Energy Backbone ($E_\phi$).** The core network takes the noisy action sequence $\boldsymbol{a}_t \in \mathbb{R}^{T_p \times D_a}$ as input.

- **Structure:** The backbone is a 1D Temporal U-Net consisting of down-sampling and up-sampling blocks with kernel size 5. Each block utilizes residual connections and Group Normalization (groups=8).

- **Conditioning:** The state embedding $\boldsymbol{c}_{state}$ and time embedding are injected into every convolutional block via Feature-wise Linear Modulation (FiLM), ensuring the energy landscape is globally conditioned on the current agent state.

**Modifications for Energy Parameterization.** To satisfy the theoretical requirement that our score field be a conservative vector field ($\nabla \times \mathcal{S} = 0$), we modify the standard Diffusion Policy architecture in two ways:

1. **Scalar Output Head:** Standard implementations output a tensor of shape $[B, T_p, D_a]$ representing the noise. We replace the final output projection. The final feature map of the U-Net (shape $[B, C, T_p]$) is aggregated via `GlobalAveragePooling1D` to capture global temporal dependencies. This is passed through a 3-layer MLP ($256 \rightarrow 128 \rightarrow 1$) to produce the single scalar energy value $E \in \mathbb{R}$.

2. $C^2$ **Differentiable Activations:** The standard ReLU activation is non-differentiable at zero. Since our training objective (Eq. (15)) involves the derivative of the score (which is the second derivative of the energy), the network must be twice-differentiable ($C^2$). We replace all ReLU activations with **Mish**. This ensures a smooth gradient flow during the double-backpropagation required for score matching.

## C.3. Differentiable Training Infrastructure

Training necessitates computing the gradient of the network output with respect to its inputs during the forward pass (to obtain the score $\nabla_{\boldsymbol{a}} E$).

**Graph Construction.** We utilize PyTorch's automatic differentiation engine. For a batch of action sequences $\boldsymbol{a}_t$ and states $\boldsymbol{s}$:

$$\mathcal{S}_\phi = -\nabla_{\boldsymbol{a}_t} E_\phi(\boldsymbol{a}_t, \boldsymbol{s}, t). \tag{42}$$

We invoke `torch.autograd.grad` with `create_graph=True`. This constructs a computational graph of the gradient operation itself, allowing the optimizer to backpropagate the Score Matching loss through the gradient computation to update the network parameters $\phi$.

**Spectral Normalization.** To encourage Lipschitz continuity, which stabilizes the energy magnitudes and prevents the "energy exploding" problem common in EBM training, we apply Spectral Normalization to the linear layers in the scalar output head.

## C.4. Hyperparameters

We train ENERGYFLOW using the AdamW optimizer with the hyperparameters detailed in Table 5.

*Table 5.* Hyperparameters for ENERGYFLOW Training and Inference.

| Parameter | Value |
|---|---|
| *Architecture* | |
| Backbone | 1D Conditional U-Net |
| Input | State Condition |
| Downsampling channels | $[64, 128, 256]$ |
| Activation Function | Mish |
| Pooling | Global Average Pooling |
| *Training* | |
| Optimizer | AdamW |
| Learning Rate | $1.0 \times 10^{-4}$ |
| Weight Decay | $1.0 \times 10^{-6}$ |
| Batch Size | 256 |
| LR Scheduler | Cosine Decay (warmup=500 steps) |
| Gradient Clipping | Norm = 1.0 |
| Noise Schedule | Geometric |
| *Inference* | |
| ODE Solver | Euler Method |
| Steps ($K$) | 20 |
| Prediction Horizon ($T_p$) | 16 |
| Observation Horizon ($T_o$) | 2 |

## C.5. Baseline Implementation

To ensure a fair evaluation, we standardize the observation encoders across all baselines. All methods utilize the same MLP-based state encoders and temporal position embeddings described in §C.1. Unless otherwise noted, we tune the hyperparameters of each baseline using a grid search over learning rates $\{10^{-3}, 10^{-4}, 10^{-5}\}$ and batch sizes $\{128, 256\}$.

### C.5.1. AUTOREGRESSIVE AND GENERATIVE POLICIES

**LSTM-GMM (Dalal et al., 2023).** We implement the LSTM-GMM policy using a standard recurrent backbone. The network consists of a 2-layer LSTM with 256 hidden units. The output head projects the hidden state to the parameters of a Gaussian Mixture Model (GMM) with $K = 5$ components, predicting means $\mu$, scales $\sigma$, and mixing coefficients $\pi$. The model is trained via Negative Log-Likelihood (NLL) maximization. During inference, we sample actions from the GMM component with the highest probability.

**Diffusion Policy (Chi et al., 2023).** To isolate the efficacy of our energy-based formulation from architectural benefits, we implement the Diffusion Policy baseline using the exact same 1D Conditional U-Net backbone as our method (see §C.1). However, instead of a scalar energy head, the baseline retains the standard vector output head to regress the noise $\boldsymbol{\epsilon} \in \mathbb{R}^{T_p \times D_a}$.

- **Training:** We use the DDPM objective with $T = 100$ diffusion steps and a squared error loss on the noise prediction.

- **Inference:** We use the DDIM scheduler with 20 denoising steps to match the inference budget of our method.

**Flow Policy (Zhang et al., 2025b).** We parameterize the conditional policy using a RealNVP-based Normalizing Flow. The architecture consists of a sequence of 4 coupling layers. Each coupling layer uses a 2-layer MLP (256 hidden units, ReLU activations) as the scale and translation network. The base distribution is a standard isotropic Gaussian. The model is conditioned on the state embedding by concatenating it to the input of the coupling layer MLPs. Training minimizes the negative log-likelihood of the expert actions.

### C.5.2. ENERGY-BASED METHODS

**Implicit BC (IBC) (Florence et al., 2021).** We implement IBC using a discontinuous energy parameterization. The energy function is an MLP with 3 layers of 512 hidden units and ReLU activations. Unlike our method, IBC does not enforce differentiability for the inference procedure; instead, it relies on derivative-free optimization.

- **Training:** We use the InfoNCE-style loss with negative samples drawn from a uniform distribution over the action bounds.

- **Inference:** We employ the Derivative-Free Optimizer (DFO) proposed in the original paper (Autoregressive derivative-free search) to find the energy minimum.

**EBT-Policy (Davies et al., 2025).** Following the official implementation, we use a Transformer-based architecture to parameterize the energy function. The model processes the state and action sequence as tokens. We use a 4-layer Transformer Encoder with 4 attention heads and an embedding dimension of 128. The model is trained using Noise Contrastive Estimation (NCE). Inference is performed using Langevin Dynamics for $K = 100$ steps with a step size of 0.01.

### C.5.3. INVERSE REINFORCEMENT LEARNING (IRL)

For IRL baselines, which recover a reward function to train a policy, we use Soft Actor-Critic (SAC) as the underlying RL optimizer. The details can be found in Appendix C.6.1.

**IQ-Learn (Garg et al., 2021).** We implement IQ-Learn (Implicit Q-Learning), which avoids adversarial training by learning a Q-function that implicitly represents both the reward and the policy. We use a clipped double-Q architecture (MLP with sizes [256, 256]). The policy is defined as $\pi(\boldsymbol{a}|\boldsymbol{s}) \propto \exp(Q(\boldsymbol{s}, \boldsymbol{a}) - V(\boldsymbol{s}))$.

**EBIL (Liu et al., 2021).** Energy-Based Imitation Learning (EBIL) is trained using an adversarial setup. We use an MLP-based energy function $E_\psi(\boldsymbol{s}, \boldsymbol{a})$ as the discriminator/reward. The policy is optimized to maximize the cumulative energy values via SAC, while the energy function is updated to assign lower energy to expert data and higher energy to policy samples using a partition function approximation.

**NEAR (Diwan et al., 2025).** We implement NEAR with its proposed NCSN neural network with ELU activations. The NCSN noise scale was defined as a geometric sequence with $\sigma_1 = 20$, $\sigma_L = 0.01$, and $L = 50$. The exponentially moving average (EMA) of the weights of the energy network during training is tracked and used during inference to enchance stability in the sample quality.

### C.6. RL Implementation

#### C.6.1. SOFT ACTOR-CRITIC ALGORITHM

We use Soft Actor-Critic (SAC) (Haarnoja et al., 2018) as the off-policy RL optimizer in Sec 5. SAC learns a stochastic policy $\pi_\phi(\mathbf{a} \mid \mathbf{s})$ together with two action-value functions $Q_{\theta_1}(\mathbf{s}, \mathbf{a})$ and $Q_{\theta_2}(\mathbf{s}, \mathbf{a})$ (clipped double-Q) and their target networks. The policy outputs a diagonal Gaussian distribution, sampled via the reparameterization trick, followed by a $\tanh$ squashing function to enforce bounded actions; the squashed outputs are then linearly rescaled to the environment's valid ranges.

Given a data batch $(\mathbf{s}_t, \mathbf{a}_t, \mathbf{s}_{t+1}, d_t)$ with $d_t \in \{0, 1\}$ is the success indicator, SAC minimizes the soft Bellman error. Let $\mathbf{a}_{t+1} \sim \pi_\phi(\cdot \mid \mathbf{s}_{t+1})$. The target is

$$y_t = r(\mathbf{s}_t, \mathbf{a}_t, \mathbf{s}_{t+1}) + \gamma(1 - d_t) \left( \min_{i \in \{1,2\}} Q_{\bar{\theta}_i}(\mathbf{s}_{t+1}, \mathbf{a}_{t+1}) - \alpha \log \pi_\phi(\mathbf{a}_{t+1} \mid \mathbf{s}_{t+1}) \right), \quad (43)$$

where $\gamma$ is the discount factor, $\alpha$ is the entropy temperature, and $Q_{\bar{\theta}_i}$ are target critics. Each critic is updated by

$$\mathcal{L}_Q(\theta_i) = \mathbb{E}\left[ \left( Q_{\theta_i}(\mathbf{s}_t, \mathbf{a}_t) - y_t \right)^2 \right]. \quad (44)$$

The actor objective is

$$\mathcal{L}_\pi(\phi) = \mathbb{E}\left[ \alpha \log \pi_\phi(\mathbf{a}_t \mid \mathbf{s}_t) - \min_{i \in \{1,2\}} Q_{\theta_i}(\mathbf{s}_t, \mathbf{a}_t) \right], \quad (45)$$

with $\mathbf{a}_t \sim \pi_\phi(\cdot \mid \mathbf{s}_t)$. We use automatic entropy tuning with target entropy $\mathcal{H}_{\text{tgt}} = -\dim(\mathbf{a})$. Target networks are updated by Polyak averaging $\bar{\theta}_i \leftarrow \tau\theta_i + (1 - \tau)\bar{\theta}_i$.

C.6.2. EXPERIENCE REPLAY BUFFER

We use experience replay to stabilize SAC training. The replay buffer $\mathcal{D}_{\text{agent}}$ contains transitions collected under the current policy. $\mathcal{D}_{\text{agent}}$ is a FIFO buffer with a fixed capacity; once full, the oldest transitions are overwritten. We sample uniformly from $\mathcal{D}_{\text{agent}}$ for SAC actor/critic updates. We use the environment-provided termination signal and store it as $d_t$, with time-limit truncation as timeouts.

### C.7. OOD Perturbation Implementation

We adopt the out-of-distribution (OOD) perturbation protocols from Pomponi et al. (2025). The S and M perturbation levels correspond to the $D_0$ and $D_1$ datasets, respectively, which feature progressively larger perturbations to the initial positions of objects that the agent must contact. The L perturbation level ($D_3$ dataset) extends this protocol by additionally perturbing the positions of target positions or objects.

## D. Experiment Tasks

### D.1. Simulation Tasks

**RoboMimic Tasks**    In the `Can` task, the robot needs to lift a soda can from one box and put it into another box. In the `Lift` task, the robot needs to lift a cube above a certain height. In the `Square` task, the robot needs to fit the square nut onto the square peg. The `Transport` task entails the collaborative effort of two robot arms to transfer a hammer from a closed container on one table to a bin on another table. One arm is responsible for retrieving and passing the hammer, while the other arm cleans the bin and receives the passed hammer. In `Tool Hang`, the robot needs to insert the hook into the base to assemble a frame and then hang a wrench on the hook.

**Meta-World Tasks**    `ButtonPress` and `DrawerOpen` evaluate the agent's ability to interact with articulated objects, requiring the robot to apply force to a switch or manipulate a constrained joint mechanism, respectively. `BinPicking` tests robust grasping and retrieval from a confined volume. `Assembly` requires aligning a circular nut with a matching peg under tight tolerances, while `Hammer` involves tool use, where the agent must grasp a hammer and accurately drive a nail into a target surface.

## E. Additional Experiment Details

### E.1. Simulation Task Demonstration

E.1.1. ROBOMIMIC TASKS

Figure 6 illustrates the successful execution of each task on RoboMimic benchmark, with our ENERGYFLOW policy.

E.1.2. META-WORLD TASKS

Figure 7 illustrates the successful execution of each task on Meta-World benchmark, with our ENERGYFLOW policy.

### E.2. Real Robot Experiment

For each task, we collect 10 teleoperated demonstrations. Following (Chi et al., 2023), we augment training data with random crops. During inference, we take a static center crop with the same size. The policy operates at 10Hz, receiving $226 \times 226$ RGB images and outputting 8-dimensional actions (7 joint velocities + gripper command). We use a ResNet-18 encoder (He et al., 2016) pretrained on ImageNet as our visual backbone, consistent with prior work (Chi et al., 2023; Zhao et al., 2023).

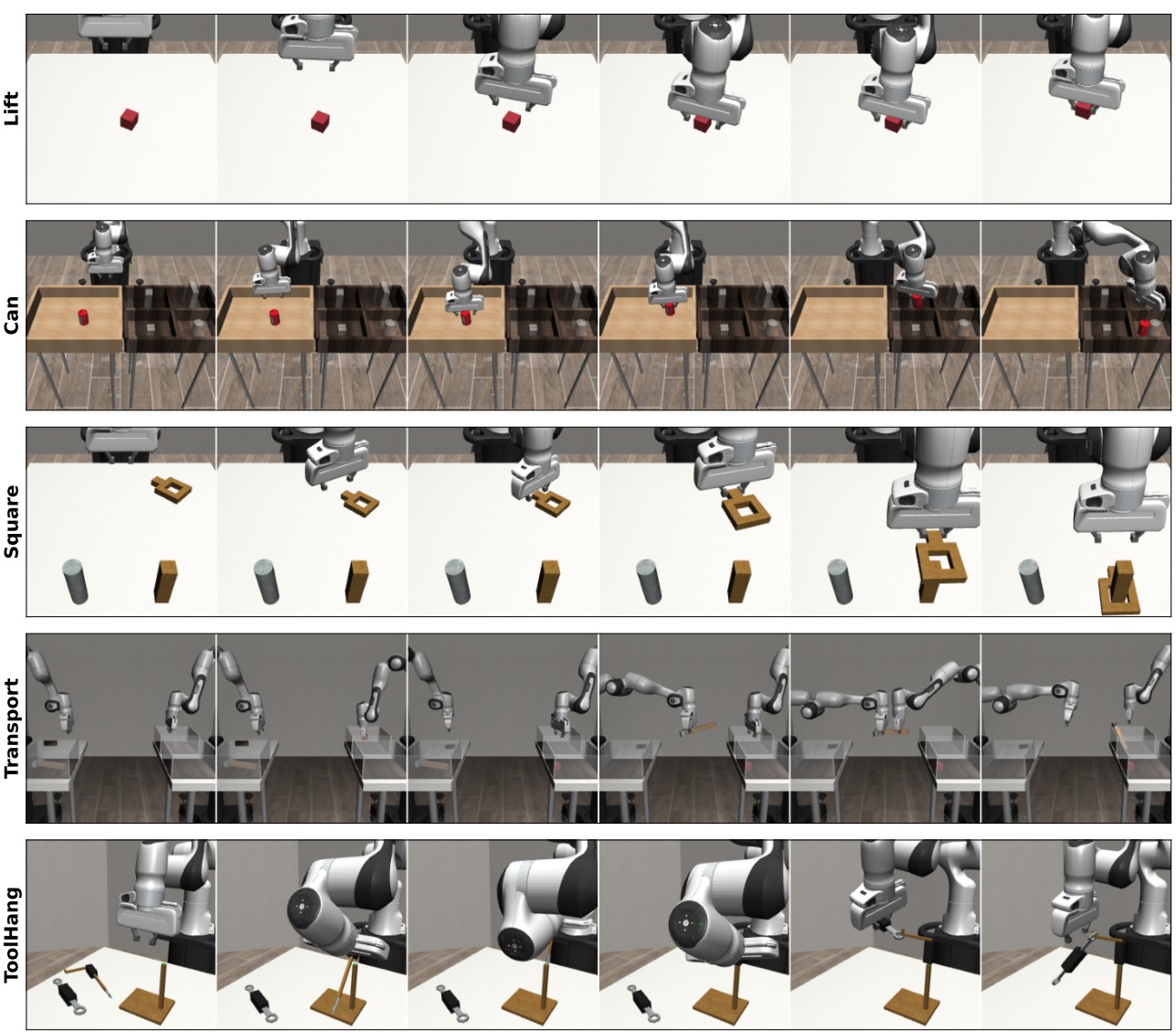

*Figure 6.* **RoboMimic task demonstrations.** Each row visualizes a rollout sequence for a different task.

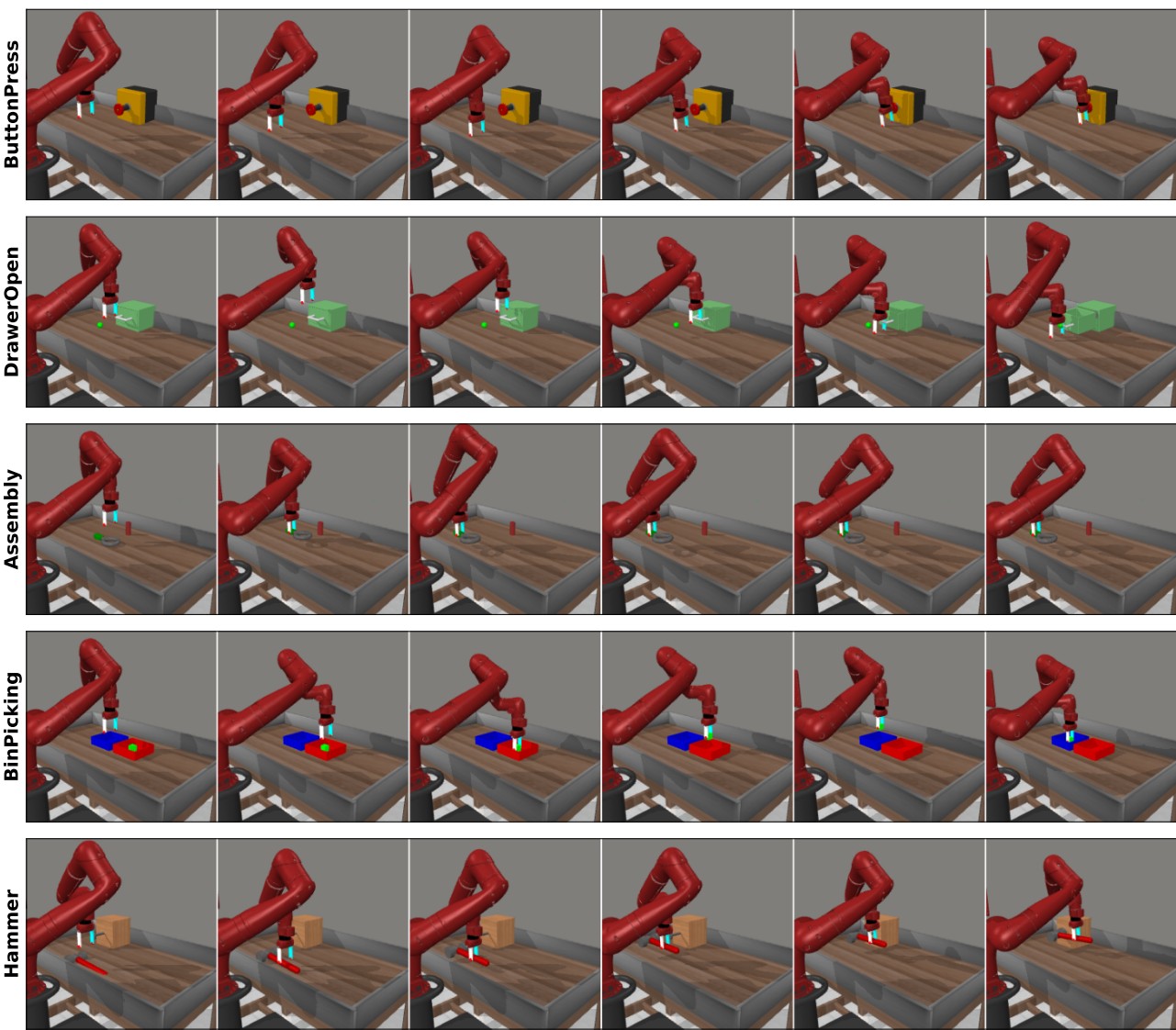

*Figure 7.* **Meta-World task demonstrations.** Meta-World task demonstrations. Each row visualizes a rollout sequence for a different task.

