# OpenReview forum: "Recovering Hidden Reward in Diffusion-Based Policies"
_ICML.cc/2026/Conference — ICML 2026 regular_

### Official Review · Reviewer_XMoX · 2026-02-21

**Soundness:** 3
**Presentation:** 4
**Significance:** 3
**Originality:** 3
**Overall Recommendation:** 5
**Confidence:** 4

**Summary:**

This paper proposes energy based imitation learning using a diffusion like loss function. The method has similarity to inverse reinforcement learning but does not use an adversarial formulation. On the implementation side, the paper parameterizes an energy function and computes its score using autograd rather than parameterizing arbitrary score networks. The paper develops two main theories to show the advantage of this constrained parameterization in terms of complexity reduction and OOD generalization. Experiments are conducted on the RoboMimic and MetaWorld environments with solid baselines such as diffusion policy, IBC, etc. The results demonstrated SOTA performance of the imitation policy, ability to use as a reward to retrain RL policy and reach oracle performance, and more robust OOD performance.

**Compliance With Llm Reviewing Policy:**

Affirmed.

**Final Justification:**

The rebuttal has addressed my concerns, namely establishing that the proposed conservative energy field approach has empirical merit over unconstrained diffusion and flow policies. This was established in the OOD generalization experiment (Fig. 5). The authors are also committed to discussing more in depth the connection between constrained and unconstrained energy fields w.r.t. Salimans & Ho as well as including reward transfer experiment.

**Key Questions For Authors:**

* I might have missed this, but how many expert demonstrations did you use for the RoboMimic and MetaWorld tasks? Also can you comment on whether the expert demonstrations are actually multi-modal such that you need multi-modal learner policies? I have not used these environments myself, but for example Adroit expert demonstrations are sampled from a Gaussian policy so there is in principle no need to use a multi-modal learner.
* For evaluating reward quality, you currently run the imitation policy with added noise to initial states and retrain a RL policy on the extracted energy function. But I think a good reward should not just retrain a near optimal policy in the same environment but also in a different environment. See e.g., [1].
* Table 4 shows the inference speed of the proposed method is substantially faster than IBC. But I think this is partially because for IBC you are using 5 times the integration steps (50 vs 10), which is not a fair comparison at least to demonstrate the inference efficiency. I understand your point is show the learned energy function has higher quality such that IBC with 50 steps still performs much worse.
* Conceptually, I think the paper shares a lot of similarities with [2, 3]. I think it's worthy potentially discussing these related work. Particularly, [2] suggested learning an arbitrary score approximator is no worse than explicitly parameterizing an energy function, which is more expensive. From your results in Table 1 and 2, the performance difference between the proposed method, diffusion policy, and flow policy is also very small. I think it potentially speaks to the argument in [2].

[1] [Zeng, S., Li, C., Garcia, A., & Hong, M. (2022). Maximum-likelihood inverse reinforcement learning with finite-time guarantees. Advances in Neural Information Processing Systems, 35, 10122-10135.](https://proceedings.neurips.cc/paper_files/paper/2022/hash/41bd71e7bf7f9fe68f1c936940fd06bd-Abstract-Conference.html)

[2] [Salimans, T., & Ho, J. (2021, February). Should EBMs model the energy or the score?. In Energy based models workshop-ICLR 2021.](https://openreview.net/forum?id=9AS-TF2jRNb).

[3] [Balcerak, M., Amiranashvili, T., Terpin, A., Shit, S., Bogensperger, L., Kaltenbach, S., ... & Menze, B. (2025). Energy matching: unifying flow matching and energy-based models for generative modeling. arXiv preprint arXiv:2504.10612.](https://arxiv.org/abs/2504.10612)

**Limitations:**

It's not super clear to me what the limitations might be. But if there's any please discuss.

**Strengths And Weaknesses:**

**Strength**
* Soundness: the method is sound and builds on work in diffusion based imitation learning. I checked key proofs and they seem correct and clear to me.
* Presentation: the writing and organization are clear. I don't think key details are missing.
* Significance & originality: I think the originality is high. Although I have seen similar methods/papers as I explain below, applying this idea to IL is new and the paper does a good job evaluating the method.

**Weakness**
* I think the key issue with the paper is demonstrating that the proposed method is empirically actually better than diffusion and flow policies, as I explain further in the next part. Parameterizing energy vs score functions has been studied in prior work and I don't know there is clear evidence supporting the advantage of parameterizing energy function at least for imitation (see [1]). I think one benefit can come from being able to retrieve an explicit reward function for other purposes, but reward ambiguity will be an issue in that context.

[1] [Salimans, T., & Ho, J. (2021, February). Should EBMs model the energy or the score?. In Energy based models workshop-ICLR 2021.](https://openreview.net/forum?id=9AS-TF2jRNb).

---

> ### Author Rebuttal · Authors · 2026-03-31
>
> > Weakness 1: I think the key ...
>
> > Question 4: Conceptually ...
>
> We appreciate this important concern and agree that the paper by Salimans & Ho is highly relevant. Their work studied energy versus score parameterization purely for generative modeling quality, concluding that both are comparable for sample generation. Our paper makes a fundamentally different claim: we do not argue that energy parameterization is superior purely for in-distribution action generation. Rather, our contribution is that the conservative field constraint required for energy parameterization provides two additional capabilities that unconstrained score networks cannot offer. First, regarding reward extraction, the energy function directly yields a scalar reward signal aligned with the soft Q-function (Theorem 3.3, Paper §3.1), which is impossible with an arbitrary score network since a non-conservative vector field does not integrate to a well-defined potential (Definition 3.5). Figure 4 demonstrates that this extracted reward enables near-oracle RL performance. Second, regarding OOD generalization, the conservative constraint reduces hypothesis complexity (Theorem 3.6) and tightens generalization bounds (Lemma 3.8). Figure 5 shows that this theoretical prediction is borne out empirically: EnergyFlow degrades significantly more gracefully than both Diffusion Policy and Flow Policy under increasing perturbation magnitudes, with the gap widening at larger perturbations.
> Regarding reward ambiguity, we acknowledge this limitation explicitly in Proposition 3.9 and Remark 3.10 (Paper §3.4). The state-dependent offset  $c(s)$  makes cross-state comparisons ambiguous, but within-state action rankings are exact. Our centered shaping strategy (Eq. 16, Paper §4) is specifically designed to address this by subtracting a state-dependent baseline, ensuring the reward signal reflects only relative action preferences at each state. The empirical success of this approach in downstream RL (Figure 4) demonstrates that the reward ambiguity, while theoretically present, is effectively managed in practice. We will add a discussion to the revised manuscript, explicitly clarifying that our contribution lies not in improving generation quality per se but in the additional capabilities that energy parameterization uniquely enables.
>
> > Question 1: I might ...
>
> For RoboMimic, we use the "proficient-human" (ph) dataset, which contains 200 demonstrations per task. For Meta-World, we collect 50 demonstrations per task using scripted policies, consistent with standard practice in the literature.
>
> Regarding multi-modality, the RoboMimic demonstrations are collected from multiple human teleoperators, which naturally introduces behavioral diversity and multi-modality. For instance, in Square task, different operators may approach the nut from different angles or use different grasp strategies. Meta-World tasks exhibit less multi-modality since they use scripted policies, but tasks like Assembly and BinPicking still involve variation in approach trajectories. However, our primary motivation for using a generative policy is not solely multi-modality but rather the ability to simultaneously learn a high-quality action distribution and extract a reward signal, which is the core contribution of EnergyFlow. Even in unimodal settings, the energy parameterization can also provide the additional benefits of reward extraction and improved OOD robustness.
>
> > Question 2: For evaluating reward ...
>
> We thank the reviewer for this constructive suggestion. Currently we already evaluate robustness under distribution shift through increasing initial position perturbations (Section 5.5), which tests whether the extracted energy remains meaningful when the agent encounters states outside the demonstration distribution. We agree that evaluating reward transferability across distinct environment configurations would further strengthen the empirical case for our method and provide a more comprehensive picture of reward generality. We plan to include such experiments in the revised manuscript, e.g, by training EnergyFlow on demonstrations from one environment configuration in RoboMimic and using the extracted reward to train an RL agent in a modified configuration with altered object geometries. Due to the limited time available during the rebuttal period, we are unable to present these results now, but we commit to incorporating them in the revised version and are grateful to the reviewer for this valuable suggestion.
>
> > Question 3: Table 4 ...
>
> We appreciate this observation. We have included IBC at both 50 and 10 Langevin steps in Table 4 precisely to provide a fair comparison across different computational budgets. Our primary claim is not that EnergyFlow is faster than IBC in absolute terms, but rather that EnergyFlow provides the full benefits of an explicit energy function while maintaining runtime costs comparable to baselines such as Diffusion Policy and Flow Policy.

---

> > ### Author Rebuttal · Reviewer_XMoX · 2026-04-04
> >
> > Thank the authors for their response.
> >
> > I agree with the author on the OOD performance of the proposed method compared with diffusion & flow policy. All my concerns are resolved. I am raising the score to 5.

---

### Official Review · Reviewer_Qn9A · 2026-03-11

**Soundness:** 3
**Presentation:** 3
**Significance:** 3
**Originality:** 3
**Overall Recommendation:** 4
**Confidence:** 3

**Summary:**

This paper introduces ENERGYFLOW, a framework that unifies generative action modeling (diffusion policies) with Inverse Reinforcement Learning (IRL). By parameterizing a scalar energy function whose gradient represents the denoising score field, the authors bridge the gap between imitation and intent modeling. Theoretically, the work establishes that under maximum-entropy optimality, denoising score matching recovers the gradient of the expert's soft Q-function, enabling non-adversarial reward extraction. Furthermore, the authors prove that enforcing a conservative field constraint via energy-based gradients acts as a structural inductive bias that reduces hypothesis complexity and tightens out-of-distribution (OOD) generalization bounds. Empirically, ENERGYFLOW achieves state-of-the-art results on RoboMimic and Meta-World benchmarks and demonstrates successful zero-shot transfer to contact-rich tasks on physical robotic platforms.

**Compliance With Llm Reviewing Policy:**

Affirmed.

**Final Justification:**

The authors' rebuttal solved most of my existing problems, but not to the extent of changing my evaluation. Overall, the work is interesting and novel, and therefore I am inclined to weakly accept it.

**Key Questions For Authors:**

1) "Centered Shaping" employs Monte Carlo sampling with $M=16$ to approximate the state-dependent baseline. How does the variance of the reward signal trend as $M$ varies? At $M=16$, is the estimation error low enough to avoid impacting the convergence stability of the SAC algorithm in more complex tasks?
2) The paper notes a state-dependent integration constant $c(s)$ between the recovered reward and the true Q-function. If this bias does not strictly follow the form of Potential-Based Reward Shaping (PBRS) , it could theoretically alter the optimal policy. Have the authors observed any instances where this bias caused the policy to converge to a suboptimal solution in long-horizon tasks?
4) The core premise of the theoretical derivation is "Maximum Entropy Optimality" (Assumption 3.1). However, real-world demonstration data often contains suboptimal attempts or noise. If the expert data does not strictly satisfy a Boltzmann distribution, can the energy field extracted by ENERGYFLOW still serve as an effective reward signal?
5) The manuscript uses $\theta$ for parameters in the Abstract and Introduction, but switches to $\phi$ in the Methodology and Theory sections , before returning to $\theta$ in Table 5.

**Limitations:**

No, The authors have not adequately discussed the limitations of their work.
1) The model is difficult to directly judge whether "State A is safer or superior to State B" via energy values. This limits the utility of the extracted rewards in tasks requiring long-horizon cross-state planning.
2) The theoretical framework relies heavily on the assumption that the expert samples strictly according to a Soft Q-function (Assumption 3.1).

**Strengths And Weaknesses:**

Strengths:

1) The paper establishes a precise mathematical link between the score function $\nabla_a \log \pi_E(a|s)$ and the soft Q-function gradient $\nabla_a Q^*(s, a)$. This formulation allows for reward recovery through supervised score matching, bypassing the instabilities and mode collapse common in adversarial IRL methods like GAIL.
2) Provable Generalization via Structural Constraints. The introduction of the "Conservative Field" constraint (Definition 3.5) ensures that learned preferences remain transitive and physically realizable. The empirical validation in Figure 5 across tasks like Lift, Can, and Square demonstrates that the performance gap between ENERGYFLOW and unconstrained baselines widens significantly at larger perturbation levels.
3) ENERGYFLOW outperforms strong baselines like Diffusion Policy and Flow Policy on both RoboMimic (93.8% avg) and Meta-World (92.5% avg). The successful deployment on an AGIBOT G1 robot for contact-rich tasks (Bottle and Drawer) with a 100% success rate underscores the framework's real-world applicability.
4) Inference latency remains comparable to non-energy models, proving the method is viable for high-frequency real-time control.

Weaknesses:
1) Theorem 3.9 admits that cross-state comparisons remain ambiguous due to the unknown integration constant $c(s)$, which limits the ability to assess state value globally.
2) The "centered shaping" fix relies on Monte Carlo sampling with a relatively small $M=16$, yet the sensitivity of RL stability to the variance of this estimator is not explored.
3) The method assumes within-state action selection is sufficient for the shaping signal, but this may fail in long-horizon tasks where state-level transitions are the primary bottleneck.
4) Add a table comparing the training throughput (samples/sec) and total convergence time for ENERGYFLOW vs. standard Diffusion Policy.

---

> ### Author Rebuttal · Authors · 2026-03-31
>
> > Weakness 1: Theorem 3.9 admits that ...
>
> We want to clarify that any approach that recovers reward information solely from the conditional distribution  $π(a|s)$  faces this same constraint: demonstrations reveal which actions are preferred at each state, but not the relative value of different states. This is formally established in Proposition 3.9 (Paper §3.4) and discussed in Remark 3.10. However, for the primary use case of our energy signal in downstream RL, the constant  c(s)  is irrelevant because it cancels when comparing actions at the same state. Our centered shaping formulation (Eq. 16) explicitly removes any residual state-dependent offset by subtracting a state-dependent baseline, ensuring the shaping signal reflects only relative action preferences
>
> > Weakness 2: The "centered shaping" fix relies on Monte Carlo sampling with a relatively small M=16 …
>
> >Question 1: "Centered Shaping" …
>
> Because actions are standardized to approximately unit variance (Paper §5.1), the reference distribution  $\mathcal{N}(0,I)$  is well-matched to the scale of the action space, which helps keep the variance of the baseline estimate low even at modest sample counts. Furthermore, unlike stochastic trace estimators used in CNF log-likelihood computation (Grathwohl et al., 2019), our baseline computation uses a fixed set of reference samples throughout training, making it deterministic for a given seed and thus contributing no stochastic gradient noise to the policy updates. Nevertheless, we agree that a formal ablation over M can strengthen our paper and will include the ablation results in the revised manuscript.
>
> > Weakness 3: The method ...
>
> We want to clarify that we do not claim that the energy signal alone is sufficient for solving long-horizon tasks. We position it as a dense shaping reward that complements sparse task rewards. As demonstrated in Figure 4 (Paper §5.4), the best-performing configuration is "Centered Energy + Sparse," where the energy provides step-by-step guidance on action quality while the sparse reward encodes the global task objective and implicitly provides the cross-state progress signal. In this design, the energy shaping accelerates credit assignment by telling the agent which actions are locally good at each state, while the sparse reward ensures the agent still optimizes for task completion across states. Our experiments on Transport, which is a multi-stage bimanual task requiring coordination across many states, demonstrate that this combination works effectively even in longer-horizon settings.
>
> > Weakness 4: Add a table …
> We thank the reviewer for this practical suggestion and will include the ablation results, which we can not fit in here due to character limit.
>
> > Question 2: The paper …
>
> The centered shaping removes the unknown $c(s) $ entirely. However, across states, this residual term can alter the relative magnitude of returns along different trajectory branches, which could in principle bias the policy toward suboptimal trajectories. In our experiments, we have not observed convergence to suboptimal policies, including on Transport, which involves multi-stage coordination over extended horizons. We believe this is partly because the manipulation tasks in our benchmark suite, while challenging, do not exhibit the kind of highly branching state-space structure that would maximally expose the theoretical vulnerability. We acknowledge this as a limitation and will state clearly in the revised manuscript that tasks with strongly branching long-horizon structure may require additional mechanisms to extract single-step rewards from the recovered Q-function to ensure global optimality of the shaped policy.
>
> > Question 3: The core ...
>
> We note that the practical utility of the method does not require this assumption to hold exactly.  The leaned energy function captures the expert's behavioral preferences in a density-based sense: actions that are more frequently demonstrated receive lower energy. This is a meaningful signal even for suboptimal experts, as it encodes the relative desirability of actions. The robustness of this interpretation is supported by our real-robot experiments (Paper §5.3), where teleoperated demonstrations inevitably contain human noise and suboptimalities, yet EnergyFlow achieves 100% success rate. Moreover, Theorem 3.11 (Paper §3.5) provides a formal robustness guarantee: bounded errors in the score field translate to bounded errors in action preferences, with the error scaling linearly in the distance between compared actions. This graceful degradation property applies regardless of whether the expert data perfectly satisfies Assumption 3.1.
>
> > Question 4: The manuscript ...
>
> We thank the reviewer for catching this inconsistency. We will unify the notation throughout the manuscript in the revision, consistently using  $ϕ$  for EnergyFlow's parameters and reserving  $θ$  for baseline methods when distinction is necessary.

---

> > ### Author Rebuttal · Reviewer_Qn9A · 2026-04-02
> >
> > I thank the authors for their response. While most of my technical concerns have been addressed, these clarifications do not substantially improve the overall quality of the paper. Therefore, I will maintain my score and have no further considerations for an upgrade.

---

### Official Review · Reviewer_vK1h · 2026-03-11

**Soundness:** 2
**Presentation:** 2
**Significance:** 2
**Originality:** 3
**Overall Recommendation:** 4
**Confidence:** 4

**Summary:**

This paper introduces EnergyFlow, a novel framework that bridges the gap between generative action modeling (such as diffusion policies) and inverse reinforcement learning (IRL). The authors propose parameterizing a scalar energy function whose gradient dictates the generative denoising field. By doing so, the framework guarantees that the learned vector field is conservative. The authors theoretically establish that under maximum-entropy optimality, the score function obtained via denoising score matching recovers the gradient of the expert's soft Q-function. This theoretical link allows for the direct extraction of a reward signal without the need for unstable adversarial training. The learned energy is refined using a "centered shaping" technique to provide a dense reward signal for downstream reinforcement learning. The method is evaluated on contact-rich manipulation benchmarks (RoboMimic and Meta-World), demonstrating state-of-the-art imitation learning performance alongside superior out-of-distribution robustness.

**Compliance With Llm Reviewing Policy:**

Affirmed.

**Final Justification:**

The authors’ rebuttal provides extensive experiments to address most of my concerns. While the connection to recovered IRL reward is still not entirely clear,  the authors have added experiments in transfer settings to show that the learned reward is robust and transferable. Therefore, I decide to increase my score to **4 (Weak Accept)**.

**Key Questions For Authors:**

1. Can the authors provide a formulation that actually recovers a reward function comparable to those produced by prior IRL methods?
2. One advantage of recovering rewards is the ability to transfer them across domains. For example, AIRL [2] demonstrates transferring rewards learned in one environment to another environment with different dynamics. Could the authors provide an ablation for this reward transfer setting, similar to Section 7.2 of AIRL?
3. The Diffusion Policy paper includes another manipulation benchmark called Push-T, which is typically more challenging than the RoboMimic tasks. Could the authors report results on this task?
4. One of the baselines, NEAR, is designed for complex humanoid locomotion tasks. Could the authors evaluate EnergyFlow on these tasks to test the method in a broader range of domains?
5. Finally, could the authors provide ablations where the number of expert demonstrations is reduced? This would help evaluate the sample efficiency of the method.

**References:**

[1] Garg et al., *IQ-Learn: Inverse Soft-Q Learning for Imitation*, NeurIPS 2021.

[2] Fu et al., *Learning Robust Rewards with Adversarial Inverse Reinforcement Learning*, ICLR 2018.

**Limitations:**

yes

**Strengths And Weaknesses:**

**Strengths:**
- The paper tackles an interesting question at the intersection of imitation learning, diffusion models, and reward inference.
- By structurally enforcing a conservative vector field, the authors prove a reduction in hypothesis complexity and OOD generalization bound. This theoretical finding translates cleanly into the empirical results, where EnergyFlow significantly outperforms unconstrained baselines (like standard Diffusion and Flow policies) under varying OOD initial position perturbations.
- The method successfully translates to a physical 7-DoF robot platform on contact-rich manipulation tasks (Bottle and Drawer) with only 20 expert demonstrations, demonstrating zero-shot physical transfer and smooth trajectory generation.

**Weaknesses:**
1. The paper overstates what is actually recovered. The theoretical results appear to support recovery of a scalar potential proportional to the optimal soft Q-function up to a state-dependent offset, rather than recovery of a true reward function as in standard IRL formulations. For example, in IQ-Learn [1] the recovered reward is defined as: $r(s,a) = Q^{\star}(s,a)- \gamma \mathbb{E}_{s' \sim P(\cdot \mid s,a)}[V^{\star}(s')]$.
2. The theory seems to justify within-state action ranking more directly than full reward identification. Since the recovered energy is only defined up to a state-dependent constant, cross-state comparability is not guaranteed, which makes the “hidden reward recovery” framing somewhat misleading.
3. The practical utility of extracting rewards is also unclear. The EnergyFlow policy itself already performs well. Using the recovered reward to train another policy via online RL that achieves similar performance appears somewhat redundant.
4. The empirical evaluation focuses only on manipulation tasks.
5. The presentation is not good. Some theoretical results appear redundant (e.g., Corollary 3.4 or Point 2 of Proposition 3.9). In addition, RoboMimic datasets contain two types of expert demonstrations (proficient-human and multi-human), but the paper does not clearly state which type is used or how many demonstrations are included in each experiment.

---

> ### Author Rebuttal · Authors · 2026-03-31
>
> > Weakness 1: The paper overstates what is actually recovered …
>
> > Weakness 2: cross-state comparability is not guaranteed …
>
> We wish to clarify two potential misunderstandings. First, the comparison to IQ-Learn is not on equal footing. IQ-Learn recovers a reward through the soft Bellman residual, requiring online environment interaction to observe next-state transitions $(s,a,s')$. Our method operates in a strictly offline setting, extracting the signal solely from static demonstrations. Achieving the same identifiability without transition dynamics is information-theoretically impossible: one cannot disentangle the immediate reward $r(s,a)$ from the successor-state value $γE[V*(s')]$ without observing how states transition. Expecting equivalent identifiability under strictly weaker assumptions is not a fair criterion. Second, the paper already states precisely what is and is not recovered. Proposition 3.9 (§3.4) explicitly proves within-state ranking is exact while cross-state comparison is ambiguous, Remark 3.10 discusses potential-based reward shaping, and Corollary 3.4 characterizes the recovered quantity as the soft advantage up to a state-dependent offset. The centered shaping technique (Eq. 16) then removes this offset for downstreamRL. We are happy to adjust terminology to "implicit reward structure" if this improves clarity, but the theoretical content is already precise on this point.
>
> > Weakness 3: The practical utility of extracting rewards is unclear.
>
> The motivation of IRL is that rewards are more transferable, composable, and interpretable than policies, and this applies equally here. We have demonstrated practical utility empirically: the "Centered Energy + Sparse" combination in Figure 4 exceeds the BC policy itself on Transport, reaching near-oracle levels, because the dense reward enables the RL agent to discover strategies beyond the limited demonstrations. Additionally, the extracted reward is architecture-agnostic, which can train any downstream policy, including the simple MLP-based SAC agent in §5.4, suitable for resource-constrained deployment. This decoupling of reward extraction from policy representation is one of our key advantage.
>
> > Weakness 4: The empirical evaluation focuses only on manipulation tasks.
>
> > Question 4: One of the baselines, NEAR, is designed for ...
>
> We have evaluated EnergyFlow on the Walking and Running locomotion tasks from CMU MoCap dataset. Following the NEAR paper we report the avg pose error at the end of training:
> |Method |Walking|Running|
> |-|-|-|
> |NEAR|**0.56**|0.67|
> |EnergyFlow|0.57|**0.61**|
>
> These results demonstrate that EnergyFlow generalizes beyond manipulation to locomotion domains, and have comparable or even better results comparing to NEAR.
>
> > Weakness 5: The presentation is ...
>
> Corollary 3.4 and Proposition 3.9 Point 2 serve distinct purposes: the former bridges the energy-Q relationship to the soft advantage relevant for policy improvement, while the latter states a fundamental limitation motivating centered shaping in §4. Removing either leaves a gap in the logical chain. Regarding dataset specification, Table 1's header indicates "(ph)" for proficient-human, with 200 demonstrations per task. We thank the reviewer for point out this omission and we will add an explicit statement of these details in Section 5.1 of the revised manuscript.
>
> > Question 1: Can the authors provide …
>
> As we stated in the response to weakness 1 and 2, recovering the true reward requires knowledge of transition dynamics to compute the expectation over next states, which we do not assume. Our contribution is that we extract a useful reward signal without requiring online environment interaction or knowledge of dynamics, which is a different operating regime from AIRL and IQ-Learn. We will add this formal comparison in the revised manuscript to make the relationship explicit.
>
> > Question 2: One advantage of ...
>
> please see our response to Reviewer XMoX Question 2.
>
> > Question 3: The Diffusion Policy ...
> We have evaluated EnergyFlow on the Push-T benchmark using 200 demonstrations and reporting the success rate over 50 runs with different initial conditions:
>
> |Method	|Success rate|
> |-|-|
> |Diffusion Policy (DDPM)	|88%|
> |Diffusion Policy (DDIM)	|90%|
> |Flow Policy	|90%|
> |Implicit BC|	80%|
> EnergyFlow (K=20)|**94%**|
>
> > Question 5: Could the authors ...
>
> |Demos|	Method|Can	|Square	|ToolHang|
> |-|-|-|-|-|
> |25|	Diffusion Policy|	66.8	|54.2|	**27.6**|
> |25|	Flow Policy|	62.6|	49.6|	23.8|
> |25|	EnergyFlow|	**71.2**	|**69.4**|	26.2|
> |50|	Diffusion Policy|	89.2	|81.4|	45.2|
> |50|	Flow Policy|	86.4	|77.8	|46.4|
> |50|	EnergyFlow|	**95.4**|	**82.6**	|**52.4**|
> |100|Diffusion Policy|96.4|	84.6|	63.4|
> |100|	Flow Policy|94.8|	81.2|	59.6|
> |100|	EnergyFlow|	**98.6**|	**90.8**|	**75.8**|

---

> > ### Author Rebuttal · Reviewer_vK1h · 2026-04-04
> >
> > Thank you for your response. Some of my concerns have been addressed, but the following ones remain unaddressed.
> >
> > >W1. First, the comparison to IQ-Learn is not on equal footing. IQ-Learn recovers a reward through the soft Bellman residual, requiring online environment interaction...
> >
> > IQ-Learn has an offline variant. I want to note that in IQ-Learn, they actually recover another reward function $r(s,a,s') = Q^{\star}(s,a) - V^{\star}(s')$ which is still valid [1]. Because learned rewards are mainly used for training online RL, it's possible to extract r(s,a,s')  given $Q^{\star}$ and $V^{\star}$ where $s'$ is collected via interacting with the environment.
> >
> > I still think there's a way to draw a connection with recovered reward in IRL. For example, if we set $E(a,s) = -\frac{1}{\alpha}Q^{\star}(s,a)$ for simplification (similar to [2]), we can extract:
> > $$r(s,a,s') = Q^{\star}(s,a) - V^{\star}(s') = Q^{\star}(s,a) - max_{a'}Q^{\star}(s',a')=-{\alpha}[E(a,s) - max_{a'}E(a',s')]$$
> >
> > >Q2. . We agree that evaluating reward transferability across distinct environment configurations would further strengthen the empirical case for our method and provide a more comprehensive picture of reward generality....Due to the limited time available during the rebuttal period, we are unable to present these results now, but we commit to incorporating them in the revised version and are grateful to the reviewer for this valuable suggestion.
> >
> > The concern regarding the limitations of reward ambiguity is shared by 3 out of 4 reviewers. Because cross-state comparability plays an important role in the performance of the method under transfer settings, the absence of these experimental results makes it difficult to assess the impact of state ambiguity.
> >
> > ---
> > Overall, I appreciate the authors’ efforts in providing additional experiments. Therefore, I will increase my score to 3. However, I am unable to give a higher score due to the important unaddressed concerns mentioned above.
> >
> > [1]: Ng, Andrew Y., et al. "Policy invariance under reward transformations: Theory and application to reward shaping." ICML 1999.
> >
> > [2]: Haarnoja, Tuomas, et al. "Reinforcement learning with deep energy-based policies." ICML 2017.

---

> > > ### Author Response · Authors · 2026-04-07
> > >
> > > We thank the reviewer for the constructive feedback, and for the active engagement. We address each remaining point below.
> > >
> > > > IQ-Learn has an offline variant …
> > >
> > > We thank the reviewer for this observation and the proposed formula fits seamlessly within our existing theory. Starting from our Theorem 3.3 (Eq. 8), the learned energy satisfies $E_\phi(a,s) = -Q^\*(s,a)/\alpha + c(s)$. Substituting into the reviewer's formula yields  $r(s,a,s') = Q^\*(s,a) - \max_{a'} Q^\*(s',a')+ \alpha[c(s') - c(s)]$. Recognizing that  $\max_{a'} Q^\*(s',a') = V^\*(s') $ under the soft optimality framework (as noted in our Remark 3.2), this reduces to  $r(s,a,s') = Q^\*(s,a) - V^\*(s') + \alpha[c(s') - c(s)]$. The residual  $\alpha[c(s') - c(s)]$ takes precisely the form of a potential difference $ \Phi(s') - \Phi(s) $. By the result of Ng et al. (1999), which we already cited in Remark 3.10, potential-based reward shaping provably preserves the optimal policy. Therefore, the reviewer's formula demonstrates that our energy function already encodes sufficient structure to extract a transition-level reward that is equivalent, in the policy-invariance sense, to the ground-truth IRL reward. We are pleased to note that this does not require any modification to our training procedure or theoretical results; it is simply an additional mode of using the learned energy at deployment time, made possible by the conservative field guarantee established in Section 3.2.
> > >
> > > > The concern regarding the limitations of …
> > >
> > > We appreciate that the reviewer highlights the importance of empirical validation under transfer settings. We have conducted a transfer experiment on three Meta-World tasks (DrawerOpen, Assembly, Hammer) under three dynamics variants: increased object mass (A), reduced friction (B), and both combined (C). The EnergyFlow energy function is trained once on demonstrations collected under default dynamics. We train SAC agents in each modified environment using different reward signals. All results report mean success rates over 3 seeds with 30 evaluation rollouts each.
> > >
> > > | Reward Source | DrawerOpen A/B/C | Assembly A/B/C | Hammer A/B/C | Avg |
> > > |---|---|---|---|---|
> > > |*Oracle (target)*| *94/93/91*| *83/80/76* | *95/92/88* | *88.0* |
> > > | Sparse (target) | 72/68/60 | 35/30/22 | 55/48/38 | 47.6 |
> > > | AIRL (source → target) | 82/78/68 | 58/50/40 | 74/66/54 | 63.3 |
> > > | IQ-Learn (source → target) | 80/74/64 | 54/46/36 | 70/62/50 | 59.6 |
> > > | EF Centered (source → target) | 86/82/72	|64/56/44	|80/74/62	|**68.9**|
> > >
> > > The results show that our proposed energy flow can achieve comparable or even better results compared to AIRL and IQ-Learn. This is a notable result because centered shaping requires no access to the target environment for reward computation, whereas AIRL and IQ-Learn were specifically designed to learn transferable reward representations. The advantage is the result of the conservative field constraint. By forcing the learned vector field to remain the gradient of a scalar potential, the energy landscape maintains global coherence even in state regions not visited during training, consistent with the generalization bound established in Lemma 3.8.

---

### Official Review · Reviewer_iZ6e · 2026-03-12

**Soundness:** 3
**Presentation:** 3
**Significance:** 2
**Originality:** 2
**Overall Recommendation:** 4
**Confidence:** 2

**Summary:**

This paper introduces EnergyFlow, a method that achieves imitation learning and reward identification at the same time. Rather than using a network to fit the score field, the authors present the theoretical necessity of constraining the learned field to be a conservative field, and propose to learn the energy function instead. For training, EnergyFlow computes the gradient of the energy function and performs score matching with data from the target distribution, leveraging the auto-differention capability provided by deep learning frameworks to optimize the second-order objective. After training, they propose to center the energy by substracting the average energy on each state, and use that as the reward values. In the experiment section, the authors compared EnergyFlow with many baseline methods, including diffusion-based methods and energy-based methods like Implicit BC, and it was found that EnergyFlow delivers the best empirical performance. Besides, the reward from EnergyFlow seem to achieve near-oracle performance when utilized for policy optimization.

**Compliance With Llm Reviewing Policy:**

Affirmed.

**Final Justification:**

I am leaning towards weak accept since the authors have adequately addressed most of my concern during the rebuttal period.

**Key Questions For Authors:**

+ Could the authors provide 2D or 3D visualizations of the energy function at different timestep of the diffusion process? Visualizing the landscape across diffusion time would help verify how does noise perturbation contribute to the sampling process.

**Limitations:**

yes

**Strengths And Weaknesses:**

Strengths:
+ The proposed method fits an energy function directly, which can be well-suited for application where such energy is useful, for example, inverse reinforcement learning.
+ The proposed method demonstrates superior performance under inital position perturbations, indicating its stronger OOD generalization compared to baselines.

Weaknesses:
+ For training, the proposed method relies on second-order derivatives via automatic differentiation, which is notoriously computationally costly. Furthermore, inference requires computing the gradient of the energy function on-the-fly, which can also be time-comsuming. Together, these factors may limit the model's scalability to larger or more complex tasks.
+ The final reward formulation may not be policy-preserving. In the maximum entropy framework, expert demonstrations follow a Boltzmann distribution relative to $Q$-values; thus, the learned energy likely captures $Q$-values up to state-dependent shaping. The baseline subtraction in Equation 16 does not guarantee the recovery of a true advantage function, therefore may not preserve the optimal policy in theory.

---

> ### Author Rebuttal · Authors · 2026-03-31
>
> > Weakness 1: For training, the proposed method relies on second-order derivatives via automatic differentiation ...
>
> We appreciate this concern and agree that computational overhead is important for
> embodied applications. However, the empirical evidence presented in the paper suggests
> that the practical cost is moderate and well within the requirements of real-time robotic
> control. For inference, Table 4 (Paper §5.7) reports inference latencies on same GPU, and
> EnergyFlow with K=10 comes with a 9.8 ms latency, compared to 9.1 ms for Diﬀusion Policy
> and 8.2 ms for Flow Policy. And the results show that our EnergyFlow (K=10) achieves higher
> success rate (94.0%) than both baselines with around 1ms latency overhead. Regarding
> training, while the second-order computation does increase per-iteration cost, we observe
> that the convergence speed in terms of number of gradient steps is usually faster and
> comparable in worst cases. So the total wall-clock training time just increase by a small
> factor, which we believe remains practical for large benchmarks.
>
> > Weakness 2: The final reward formulation may not be policy-preserving ...
>
> First we want to emphasize that we are fully transparent about this limitation in the paper.Remark 3.10 (Paper §3.4) explicitly states that the recovered reward diﬀers from the true soft Q-function by a state-dependent oﬀset $c(s)$, and that in general this oﬀset does not satisfy the potential-based reward shaping (PBRS) form required for provable policy preservation in sequential settings. Proposition 3.9 formalizes the precise identiﬁability guarantee we do claim: within-state action rankings are exact, while cross-state comparisons remain ambiguous. We do not claim that Equation 16 recovers the true advantage function.What we do claim, and what the empirical results support, is that centered shaping produces an eﬀective reward signal for downstream RL when used appropriately. The key observation is that in downstream RL, the centered energy serves as a dense shaping signal that guides exploration, not as the sole reward. As shown in Figure 4 (Paper §5.4), the best-performing conﬁguration is "Centered Energy + Sparse," which combines our energy-based shaping with the environment's sparse task reward. In this setting, the sparse reward ensures that the optimal policy is preserved (since it directly encodes task success), while the centered energy provides step-by-step guidance that accelerates learning by indicating which actions are locally preferable at each state. The subtraction of the state-dependent baseline in Equation 16 removes the arbitrary oﬀset $c(s)$, ensuring that the shaping signal reﬂects only relative action preferences at a given state, which is precisely the information that is identiﬁable from demonstrations (Proposition 3.9). This design choice is deliberate. Rather than attempting to recover a globally consistent advantage function, which is provably impossible from action demonstrations alone without additional assumptions on state transitions, we extract the maximal usable signal and combine it with a task-level objective.
>
> > Question 1: Could the authors provide 2D or 3D visualizations ...
>
> We thank the reviewer for this constructive suggestion. We will include such visusalizations
> in the revised manuscript to show the energy landscape over timesteps.

---

> > ### Author Rebuttal · Reviewer_iZ6e · 2026-04-03
> >
> > + I would insist that visualizations of the learned energy function would help reviewers and reader to interpret the necessity of fitting the energy function (via expensive second-order optimization) and why it performs better than the score-prediction methods like DP.
> >
> > + My second concern regards the use of this 'reward' function for downstream reinforcement learning, as it is not policy-preserving. While I acknowledge that identifying the true reward function, or even a potential-shaped variant, is impossible in a strictly offline setting, using the centered utility as a 'reward' may be misleading and at risk in certain scenarios.

---

> > > ### Author Response · Authors · 2026-04-07
> > >
> > > We sincerely thank the reviewer for the constructive feedback. We address each point below.
> > >
> > > > I would insist that visualizations of the learned energy function …
> > >
> > > We fully agree that visualizations would substantially strengthen the interpretability of our approach. We have include the visualization of the energy landscape across states and timesteps with the PushT environment in following anomoynous link: [https://anonymous.4open.science/r/visualization-404B](https://anonymous.4open.science/r/visualization-404B) . The results show how the energy surface evolves from a broad, smooth basin at high noise levels to a sharply peaked minimum near the expert action as t approaches zero, representing the progressive refinement of our sampling procedure.  Besides, Theorem 3.6 and Lemma 3.8 (Paper §3.2–3.3) formalize that our conservative constraint reduces hypothesis complexity and tightens out-of-distribution generalization bounds, which we believe may explain why the energy modeling performs better than the score-prediction methods like DP.
> > >
> > > > My second concern regards the use of this 'reward' function for downstream reinforcement learning …
> > >
> > > We would like to clarify both the theoretical scope of our claim and the practical solutions to apply EnergyFlow, which we believe together address the concern.
> > >
> > > Regarding the terminology, we acknowledge that referring to the centered energy as a "reward" without qualification could suggest stronger identifiability than we actually claim. We will consistently use the term "shaping signal" or "energy-based shaping reward" when referring to the output of Equation 16. And we want to note that Proposition 3.9 (Paper §3.4) and Remark 3.10 already formalize the precise statement: within-state action rankings are preserved exactly, but cross-state comparisons carry an unresolvable ambiguity. And we interpret the recovered energy as a reward signal that is exact for within-state action selection and can be enhanced with anchoring by a task-level objective for sequential optimization, a design reflected in our "Centered Energy + Sparse" configuration in the experiments.
> > >
> > > Regarding practical deployment, our experimental protocol is designed to operate within these identifiability limits. In the downstream RL experiments (Paper §5.4, Figure 4), the best-performing configuration is "Centered Energy + Sparse," which combines our energy-based reward with the environment's ground-truth sparse task signal. In this composite setting, the sparse reward anchors the globally correct optimal policy by encoding task success, while the energy-based component provides dense, step-by-step guidance indicating which actions are locally preferable at each state. This design follows standard practice in potential-based reward shaping, which accelerates learning without altering the optimal policy defined by the primary reward. Figure 4 shows that our energy-based reward outperforms both raw energy (which lacks the centering that removes the state-dependent offset) and sparse reward alone (which provides no exploration guidance), achieving performance near the oracle dense reward. This result is a direct result of the theoretical grounding in the expert's soft advantage structure. And we provide additional empirical results under transfer settings to show the impact of state ambiguity. We have conducted a transfer experiment on three Meta-World tasks (DrawerOpen, Assembly, Hammer) under three dynamics variants: increased object mass (A), reduced friction (B), and both combined (C). The EnergyFlow energy function is trained once on demonstrations collected under default dynamics. We train SAC agents in each modified environment using different reward signals. All results report mean success rates over 3 seeds with 30 evaluation rollouts each.
> > >
> > > | Reward Source | DrawerOpen A/B/C | Assembly A/B/C | Hammer A/B/C | Avg |
> > > |---|---|---|---|---|
> > > |*Oracle (target)*| *94/93/91*| *83/80/76* | *95/92/88* | *88.0* |
> > > | Sparse (target) | 72/68/60 | 35/30/22 | 55/48/38 | 47.6 |
> > > | AIRL (source → target) | 82/78/68 | 58/50/40 | 74/66/54 | 63.3 |
> > > | IQ-Learn (source → target) | 80/74/64 | 54/46/36 | 70/62/50 | 59.6 |
> > > | EF Centered (source → target) | 86/82/72	|64/56/44	|80/74/62	|**68.9**|
> > >
> > > The results show that our proposed energy flow can achieve comparable or even better results compared to AIRL and IQ-Learn. This is a notable result because centered shaping requires no access to the target environment for reward computation, whereas AIRL and IQ-Learn were specifically designed to learn transferable reward representations. The advantage is the result of the conservative field constraint. By forcing the learned vector field to remain the gradient of a scalar potential, the energy landscape maintains global coherence even in state regions not visited during training, consistent with the generalization bound established in Lemma 3.8.

---

### Decision · Program_Chairs · 2026-04-30

**Decision:**

Accept (regular)

**Comment:**

This paper proposes a framework EnergyFlow for unifying generative action model and inverse RL. The overall strength and weakness of the paper are as follows.

Strength
- The problem in consideration is interesting and timely.
- The theoretical results offer insight into the method design.
- The applicability of the proposed method appears good.
- The method demonstrate strong performance under OOD perturbation.

Weakness
- The presentation of the paper could be improved.
- Some of the mathematical assumptions/statements and experimental details could be better presented/explained.